# COMET: Benchmark for Comprehensive Biological Multi-omics Evaluation Tasks and Language Models

## Abstract

As key elements within the central dogma, DNA, RNA, and proteins play crucial roles in maintaining life by guaranteeing accurate genetic expression and implementation. Although research on these molecules has profoundly impacted fields like medicine, agriculture, and industry, the diversity of machine learning approaches—from traditional statistical methods to deep learning models and large language models—poses challenges for researchers in choosing the most suitable models for specific tasks, especially for cross-omics and multi-omics tasks due to the lack of comprehensive benchmarks. To address this, we introduce the first comprehensive multi-omics benchmark COMET (Benchmark for Biological **CO**mprehensive **M**ulti-omics **E**valuation **T**asks and Language Models), designed to evaluate models across single-omics, cross-omics, and multi-omics tasks. First, we curate and develop a diverse collection of downstream tasks and datasets covering key structural and functional aspects in DNA, RNA, and proteins, including tasks that span multiple omics levels. Then, we evaluate existing foundational language models for DNA, RNA, and proteins, as well as the newly proposed multi-omics model, offering valuable insights into their performance in integrating and analyzing data from different biological modalities. We observed that DNA, RNA, and protein models can be applied to tasks across different omics by leveraging initialized embeddings, with protein models demonstrating superior performance across various omics. Through the evaluation of multi-omics tasks, we identified significant gaps in the capabilities of current models to address these challenges, highlighting substantial opportunities to enhance multi-omics integration and improve overall performance.

## 1 Introduction

Driven by curiosity about uncovering the fundamental principles of life sciences, humans have never ceased exploring the microscopic mechanisms of biological processes. DNA, RNA, and proteins, as fundamental molecules of the central dogma (Crick, 1970), play critical roles in sustaining life. Through their interrelated functions, they ensure the accurate expression and execution of genetic instructions, making them central to all biological processes. Current research on these three types of molecules has already had widespread and profound impacts across multiple fields. For example, gene sequencing and editing technologies have made early diagnosis and treatment of hereditary diseases possible (Le, 2020). Genetic modification has enabled efficient and targeted crop improvement (Ahmar et al., 2020). The analysis of protein structure and function has driven advancements in targeted drug design and the application of industrial enzymes (Śledź & Caflisch, 2018; Chapman et al., 2018). Consequently, deepening our understanding of how these molecules interact and function is crucial for unraveling the complex mechanisms underlying biological processes and driving technological and application advancements in agriculture, industry, and medicine.

Despite significant advances, current research on biological molecules faces significant challenges. With the development of machine learning technologies, research methodologies have evolved from traditional statistical approaches (Karollus et al., 2021; Bhasin et al., 2005; Kathuria et al., 2018) to deep learning models (Bogard et al., 2019; Zhuang et al., 2019; Zhang et al., 2020) and, more recently, to large language models (Chen et al., 2022; Zhou et al., 2023; Rives et al., 2021). However, this diversity of approaches has made it challenging for researchers to choose the most suitable

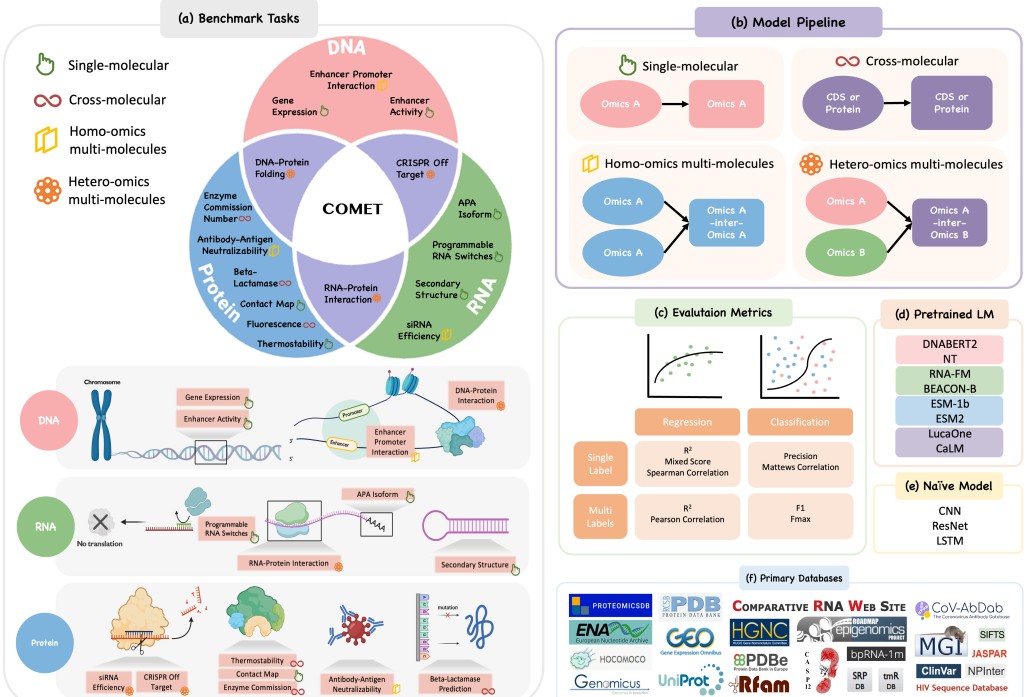

Figure 1: **Overview of COMET. (a) Benchmark Tasks:** The tasks are organized into four categories based on omics data: DNA, RNA, protein and multi-omics. They are further classified into single-molecular, cross-molecular, homo-omics multi-molecules and hetero-omics multi-molecules specified by icons to indicate the type of omics and interaction involved. **(b) Model Pipeline:** The benchmark evaluates model across four task types. Single molecular tasks have inputs and downstream task contained within a single omics type. Cross-molecular tasks utilize either CDS or protein data to perform downstream task involving the other. Homo-omics multi-molecules tasks involve two molecular interactions within the same omics type. Hetero-omics multi-molecules tasks refer to interactions spanning two omics types A.1. **(c) Evaluation Metrics:** Tasks are grouped into four by number of labels and supervised tasks types. Within each, tasks are evaluated with diverse metrics shown. **(d) Pretrained LM and (e) Naive Model:** Shows the pretrained omics language models and naive supervised models we used as baselines. **(f) Primary Databases:** Lists the primary data source we adopted and processed for our model training and evaluation.

model for their specific tasks. To address this issue, kinds of benchmarks have been established for specific scenarios. For example, benchmarks like BEND (Marin et al., 2023), BEACON (Ren et al., 2024) and PEER (Xu et al., 2022) have been developed specifically for DNA, RNA and protein-related tasks. These benchmarks serve as reference points for evaluating models in the specific domain, helping researchers compare performance across different methods and choose the most suitable approaches for their work.

Inspired by the unified models in the NLP field (Ray, 2023), scientists are now striving to develop foundational models capable of integrating different omics data in biology. This has led to growing interest in cross-omics and multi-omics research. By exploring the relationships between multi-omics data, they aim to uncover deeper insights into biological processes and foster innovation across research and application domains (Outeiral & Deane, 2024; He et al., 2024). However, as previously mentioned, the community still lacks a comprehensive benchmark for evaluating cross-omics and multi-omics tasks. Such a benchmark is essential for clearly defining the critical issues in multi-omics research and guiding its future direction.

Therefore, we propose a comprehensive benchmark called COMET (Benchmark for Biological **CO**mprehensive **M**ulti-omics **E**valuation **T**asks and Language Models) for the evaluation of single-omics, cross-omics, and multi-omics biological tasks. We select key single-omics tasks from DNA, RNA, and proteins covering structure, function and engineering, allowing for a thorough assessment of the models' performance in specific contexts. In addition, we collect the corresponding codon sequences for proteins and several downstream tasks that span multiple omics. For the models, we

select two key foundational models from each omics field, and test the newly proposed multi-omics method LucaOne (He et al., 2024). The benchmark allows for a deeper understanding of models performance in integrating and analyzing data from different biological modalities.

Through the observation of our experimental results, we surprisingly find that using models from other modalities can achieve comparable performance. For example, DNA/RNA models can perform protein tasks well. Additionally, we discovered that pretrained protein language models and multi-omics models have strong potential for multi-molecule understanding. To the best of our knowledge, we are the first to propose a comprehensive biological multi-omics benchmark, aimed at evaluating models across single-omics, cross-omics, and multi-omics tasks. The overview of COMET is shown in Figure 1. Our contributions can be summarized as follows:

- We build the first comprehensive biological multi-omics benchmark, covering key structural and functional tasks and data in single-molecules, and compiling tasks and data involving cross-molecules and multi-molecules.
- We evaluate existing foundation models for DNA, RNA, and proteins, as well as multi-omics approaches. We conduct experiments with fully-finetuned or frozen models. This provides a reference for researchers in selecting methods with appropriate finetuning ways.
- We design various approaches to evaluating cross-omics and multi-omics, bridging the gap between different omics. We find that different omics models can benefit other types of omics tasks and multi-molecular tasks still present significant challenges to face.

## 2 RELATED WORK

**Biology language models.** The rapid evolution of biology language models has significantly advanced computational biology by leveraging natural language processing techniques to analyze biological sequences. DNABERT (Ji et al., 2021), DNABERT2 (Zhou et al., 2023), the Nucleotide Transformer (Dalla-Torre et al., 2023) and Genomics-FM (Ye et al., 2024) adapt the BERT architecture for DNA sequences, but differ primarily in terms of tokenizer design, model parameters and pre-trained data. HyenaDNA (Nguyen et al., 2024) models long-range dependencies at single-nucleotide resolution. The emergence of foundational RNA models, including RNA-FM (Chen et al., 2022), BEACON-B (Ren et al., 2024) and UTR-LM (Chu et al., 2024) utilize sophisticated language modeling techniques. These models have demonstrated the ability to address a wide array of RNA-related tasks, thereby offering deeper insights into RNA biology. In protein analysis, The ESM family (Rives et al., 2021; Meier et al., 2021; Lin et al., 2023; Hayes et al., 2024) highlights the impact of scaling unsupervised learning through the use of large protein sequence datasets and a transformer-based architecture capable of capturing complex dependencies within sequences. For multi-omics models, LucaOne (He et al., 2024) and CD-GPT (Zhu et al., 2024) unify nucleic acid and protein data, while differ in training method. In the realm of cross-omics, Calm (Outeiral & Deane, 2024) introduces codon embeddings to protein language models, enhancing predictive performance by leveraging biological data containing richer signals. Boshar et al. (2024) explore the efficacy of genomic language models on protein tasks. Prakash et al. (2024) explore the possibility of applying pre-trained DNA and protein models to RNA tasks. These advancements underscore the transformative potential of large-scale language models in decoding genomic, transcriptomic, and proteomic complexities, fostering breakthroughs in molecular biology, and protein science.

**Benchmarks in biological language.** In the field of biological language processing, benchmarks have been instrumental in driving advancements across various molecular research areas. For DNA, significant contributions include Genomic Benchmarks (Grešová et al., 2023) and BEND (Marin et al., 2023), which collect diverse DNA tasks like gene finding, enhancer annotation, and CpG methylation. (Kao et al., 2024) introduces the evaluation of models for long-range tasks. RnaBench (Runge et al., 2024) primarily focuses on RNA secondary structure and design tasks. BEACON (Ren et al., 2024) is introduced to evaluate language models on RNA tasks and proposes a strong baseline model called BEACON-B. Protein benchmarks such as ProteinGym (Notin et al., 2024), tailored for protein fitness prediction and design. PEER (Xu et al., 2022), encompasses a broad range of tasks including function prediction, localization, and structure analysis. However, there is still a lack of benchmarks that comprehensively integrate multi-molecular tasks, which is crucial for capturing the complex interactions between different moleculars in biological systems. Inspired by exciting multi-omics works and the lack of corresponding benchmarks, we present the

Table 1: Overview of tasks of COMET across different molecule groups. Residue-level tasks require labels to have the same length as input nucleotide or amino acid sequences. Sequence-level tasks require one input sequence to share one label. Cls and Reg denote classification and regression.

| Task | Omics | #Train/Val/Test | Metric | Task Type | Biology Level | Max/Mean Length1/2 | | Source/Venue |
|------|-------|-----------------|--------|-----------|---------------|--------------------|---|-------------|
| **DNA** | | | | | | | | |
| GE | DNA | 16,413/1,000/1,000 | $R^2$ | Reg | Sequence | 6,000/6,000 | - | Xpresso/CR |
| EA | DNA | 402,296/40,570/41,186 | PCC | Multi-label Reg | Sequence | 249/249 | - | DeepStarr/NM |
| **RNA** | | | | | | | | |
| SSP | RNA | 10,814/1,300/1,305 | F1 | Multi-label Cls | Residue | 499/133.8 | - | SPOT-RNA/NC |
| APA | RNA | 145,463/33,170/49,755 | $R^2$ | Reg | Sequence | 186/186 | - | APARENT/Cell |
| PRS | RNA | 73,227/9,153/9,154 | $R^2$ | Multi-label Reg | Sequence | 148/148 | - | Angenent-Mari's/NC |
| **Protein** | | | | | | | | |
| Ther | Protein | 5,056/639/1,336 | SCC | Reg | Sequence | 2,694/579.5 | - | FLIP/NeurIPS |
| Cont | Protein | 25,299/224/40 | P@L | Multi-label Cls | Residue | 4,914/226.5 | - | Proteinnet/BMC Bio |
| EC | Protein | 13,090/1,465/1,604 | Fmax | Multi-label Cls | Sequence | 751/270.2 | - | DeepFRI/NMI |
| **Cross-molecules CDS/Protein** | | | | | | | | |
| Flu | CDS/Protein | 21464 / 5366 / 27217 | SCC | Reg | Sequence | 714/714 | - | Sarkisyan's/Nature |
| EC | CDS/Protein | 11105/1397/1458 | Fmax | Multi-label Cls | Sequence | 7,581/1,021.6 | - | DeepFRI/NMI |
| Beta-Lact | CDS/Protein | 9202 / 2322 / 1080 | SCC | Reg | Sequence | 858/858 | - | Firnberg's/MBE |
| **Multi-molecules** | | | | | | | | |
| EPI | DNA-DNA | 149,328/18,666/18,667 | MCC | Multi-class Cls | Sequence | 2,000/2000 | 3,000/3000 | EPI-DLMH/BIB |
| siRNA | RNA-RNA | 18,186/2,273/2,274 | Mixed Score | Reg | Sequence | 21/20.98 | 17,911/4,145.46 | SAIS/TIANCHI |
| AAN | Protien-Protien | 22,359/1,242/3,301 | MCC | Multi-class Cls | Sequence | 271/238.77 | 912/856.25 | DeepAAI/NMI |
| RPI | RNA-Protien | 14,994/1,666/4,164 | MCC | Multi-class Cls | Sequence | 3,999/1,834.7 | 3,678/523.4 | NPInterv2.0/NAR |
| CRI-Off | RNA-DNA | 14,223/2,032/4,064 | SCC | Reg | Sequence | 23/23 | 23/23 | DeepCRISPR/GB |
| DPF | DNA-Protein | 500/55/128 | LDDT | Multi-label Reg | Residue | 290/50.11 | 573/124.39 | DeepPBS/NM |

first benchmark that encompasses single-omics tasks, cross-omics tasks, and multi-omics tasks spanning DNA, RNA, and protein sequences.

## 3 BENCHMARK TASKS

The following sections provide detailed information for 17 diverse tasks, including data statistics, evaluation metrics, and data sources as shown in Table 1.

### 3.1 DNA TASK

**Gene Expression (GE)** predicts the expression levels of genes and transcription factors (TFs) across diverse tissues, recorded in target $y \in \mathbb{R}$. We extracted 1575 TF expression datasets from the GTEx database after filtering out non-expressed TFs, ensuring high-quality and tissue-specific expression profiles. Additionally, we integrated gene expression data from Xpresso (Agarwal & Shendure, 2020), covering 56 tissues, and grouped them into 11 functional and regional categories, enabling a comprehensive analysis of cross-tissue gene expression. This regression task aims to evaluate the model's ability to predict gene and TF expression levels using the $R^2$ value as the evaluation metric. *Impact:* Gene expression is crucial for understanding transcriptional regulation and functional dynamics across the genome. Accurately predicting expression levels can elucidate the regulatory networks underlying different cellular states and tissue-specific functions, offering deeper insights into how genetic and environmental factors jointly influence gene expression and contribute to phenotypic variation and disease mechanisms.

**Enhancer Activity Prediction (EA)** is a regression task that predicts enhancer activity for two promoters associated with distinct developmental and housekeeping transcriptional programs directly from the DNA sequence. The enhancer activity dataset released in (de Almeida et al., 2022) comprises 484,052 DNA sequences, each 249 nucleotides in length, measured for their quantitative enhancer activity towards either a developmental or a housekeeping promoter by a continuous target variable $y \in \mathbb{R}$. We employ PCC as the metric. *Impact:* Enhancers are essential genomic elements that regulate cell type-specific transcription of target genes, influencing animal development and physiology. Their ability to activate transcription outside their native contexts suggests that critical regulatory information resides within their DNA sequences. Mutations in enhancers can alter their function, leading to developmental defects and contributing to human diseases. Understanding enhancer activity is crucial for revealing the regulatory networks governing gene expression.

### 3.2 RNA TASK

**APA Isoform Prediction (APA)** predicts the polyA site strength for each variant, represented as target $y \in \mathbb{R}$. We filter 228k sequences from Bogard's dataset (Bogard et al., 2019) containing over

3 million APA reporter gene data . This regression task evaluates the proportion of proximal APA isoforms with the performance metric being the $R^2$ value.

*Impact:* Alternative polyadenylation is a key regulatory mechanism that diversifies RNA transcripts and protein isoforms through 3' UTR processing. By influencing transcription termination and interacting with RNA splicing, APA modulates gene expression, impacting various cellular functions.

**Programmable RNA Switches (PRS)** are synthetic RNA molecules designed to regulate gene expression by responding to specific RNA sequences. Each switch exists in one of three activity states, ON, OFF, or ON/OFF, depending on the presence or absence of its trigger RNA sequence. The target $y \in \mathbb{R}^3$ represents the three activity states. The dataset (Angenent-Mari et al., 2020) includes 91,534 in vivo toehold switches, covering 23 viral genomes and 906 human transcription factors. The activity of these switches is measured using GFP signal intensity, which provides a quantitative readout of switch performance. The effectiveness of these switches is evaluated using the $R^2$ metric.

*Impact:* Programmable RNA switches provide precise control of gene expression and cellular functions, making them essential in synthetic biology for manipulating biological processes both in vitro and in vivo. These RNAs act as responsive elements to small molecules, proteins, or nucleic acids, allowing fine regulation of cellular behavior. Therapeutically, they promise to enable targeted treatments by detecting disease-specific signals and triggering precise cellular responses, contributing to novel approaches in precision medicine and synthetic biology.

**Secondary Structure Prediction (SSP)** identifies paired nucleotide regions in stems and unpaired nucleotide regions in loops, bulges and junctions within RNA molecules. In a RNA molecule with a length of $l$, The target matrix $y \in \mathbb{R}^{l \times l}$ indicates whether each nucleotide and other nucleotides form a base pair. We utilize the bpRNA-1m database (Danaee et al., 2018). The performance metric for this task is the F1 score.

*Impact:* Accurate secondary structure prediction is paramount for elucidating the intricate mechanisms underlying function and dynamics. By mapping these structures, researchers gain insights that advance genetic research and guide RNA-based therapeutic development, supporting innovations in precision medicine.

### 3.3 PROTEIN TASK

**Thermostability Prediction (Ther)** is a regression task that aims to predict the stability of proteins at high temperatures. From the Thermostability task of FLIP (Dallago et al., 2021), we apply the 'Human-cell' splits. We use the Spearman correlation coefficient (SCC) as the metric.

*Impact:* Protein thermostability prediction advances our understanding of protein functions and properties. In industrial enzyme applications, developing highly thermostable enzymes is essential for operating under harsh reaction conditions (Wu et al., 2023). Such predictions facilitate directed evolution and selection of proteins, which is of great significance for drug and vaccine discovery (Chen & Gong, 2022).

**Enzyme Commission Number Prediction (EC)** involves annotating protein sequences. The data is sourced from the EC benchmark established in DEEPFRI (Gligorijević et al., 2021). We collect the corresponding codon sequences for the dataset simultaneously. It retains over 90% of the original samples after filtering out anomalous data. We use Fmax as the metric.

*Impact:* Identifying enzymes and their catalytic reaction types is crucial for understanding enzyme functions. This can accelerate the discovery of new enzymatic activities and improve the functions of existing enzymes. In the field of drug discovery, it can assist in designing enzymes with specific catalytic activities, supporting the development of novel drugs and therapies (Chautard et al., 2009).

**Contact Map Prediction (Cont)** aims to forecast interactions between amino acid residues in a protein. Given a protein's amino acids sequence, the target is to predict which residues are in close proximity in its three-dimensional structure. The data is sourced from the Proteinnet (AlQuraishi, 2019) and TAPE benchmark (Rao et al., 2019). We use P@L/5 as the metric.

*Impact:* Accurate contact map prediction has a significant impact on the field of structural biology (Vendruscolo et al., 1997). It helps identify protein-protein interactions, protein folding pathways, and design new proteins with desired properties. As a powerful tool, it bridges the gap between protein sequence and structure.

**Fluorescence Prediction (Flu)** evaluates the model's ability to predict fluorescence values for higher-order mutated green fluorescent protein (avGFP) sequences. The original data comes from

Sarkisyan et al. (2016) and Xu et al. (2022), following the settings outlined in Boshar et al. (2024). It's a regression task and we use SCC as the metric.

*Impact:* Accurately predicting the fluorescence values of higher-order mutated avGFP is crucial for understanding protein function. It helps to elucidate the relationship between sequence variations and function, enhancing the understanding of evolutionary landscapes. This enables the design of novel fluorescent proteins with specific properties, applicable in broader scenarios within synthetic biology.

**Beta-Lactamase Prediction (Beta-Lac)** aims to explore the fitness landscape of all single-codon mutations in a gene. The labels indicate the ability of mutated genes to confer resistance to penicillin. The data is sourced from Firnberg et al. (2014), and the complete settings from Boshar et al. (2024) are adopted. We use SCC as the metric.

*Impact:* Accurately predicting the impact of mutations on enzyme activity aids in understanding evolutionary processes at the molecular level and the mechanisms of disease. By predicting mutation effects, the specific functional mechanisms of proteins can be revealed. This has broad applications in drug development, disease treatment, industrial production, and agricultural production.

### 3.4 MULTI-MOLECULAR TASK

**Enhancer-Promoter Interaction Prediction (EPI)** is a single-label classification task in genomics that aims to identify interactions between enhancers and promoters sequence, with the categorical label $y \in \{0, 1\}$. The dataset, sourced from EPI-DLMH (Min et al., 2021), comprises six cell lines—GM12878, HUVEC, HeLa-S3, IMR90, K562, and NHEK. We sample to balance true EPIs and non-EPIs. We use the Matthews Correlation Coefficient (MCC) as the metric.

*Impact:* Enhancers are regulatory elements that can significantly enhance the transcription of genes located at varying distances, while promoters are essential regions where transcription begins. Understanding these interactions is crucial for deciphering the complex regulatory networks that govern cellular functions and can provide insights into developmental biology and disease mechanisms.

**siRNA Efficiency Prediction (siRNA)** is a regression task that aims to predict the silencing efficiency of different siRNAs. By inputting artificially modified siRNA sequences the target mRNA sequence, the model can estimate how effectively each siRNA silences its corresponding mRNA. This is a regression task, and we use Mixed Score as the evaluation metric A.5.1. The data utilized in this research are from SAIS (SAIS, 2020).

*Impact:* RNA interference (RNAi) is a natural gene expression regulation mechanism that reduces target protein levels by inhibiting the expression of target genes, typically achieved through siRNAs (Setten et al., 2019). With the success of mRNA vaccines in COVID-19 prevention, there has been growing interest in the development of nucleic acid-based drugs. Predicting the silencing efficiency of chemically modified siRNA sequences under the RNAi mechanism is crucial, as this metric is directly linked to the actual therapeutic efficacy of the drug.

**Antibody-Antigen Neutralizability Prediction (AAN)** is used to assess whether there is an interaction between an antigen and an antibody, making it a single-label classification task with the categorical label $y \in \{0, 1\}$. The goal is to predict whether a given antibody can bind to a specific antigen based on their sequences. This task is based on the HIV data from CATNAP (Yoon et al., 2015) and DeepAAI (Zhang et al., 2022). We use MCC as the metric.

*Impact:* To demonstrate the neutralizing effects of most natural and synthetic antibodies against any antigen, time-consuming, labor-intensive, and costly wet lab experiments are typically required (Lee et al., 2007). However, with machine learning, we can represent the neutralizing response of antibodies (Ab) from the perspective of Ab-Ag neutralization effects, highlighting similarities in binding regions. This approach also helps to enable the recommendation of broad-spectrum antibodies that can target new viral variants.

**RNA-Protein Interaction Prediction (RPI)** aims to forecast whether a non-coding RNA (ncRNA) interacts with RNA-binding protein (RBP). The dataset is sourced from NPInterv2.0 (Yuan et al., 2014). Since the databases only provide positive samples, which are pairs of ncRNA and proteins that interact, we utilized a negative sample dataset generated by ncRPI-LGAT (Han & Zhang, 2023), where ncRNAs and proteins are randomly paired. We use MCC as the metric.

*Impact:* The interaction between ncRNAs and RBPs plays a crucial role in various important biological processes, such as gene expression, chromatin modification, and epigenetic regulation. However, identifying ncRNA-protein interactions through wet-lab experiments remains time-consuming and expensive. The development of computational approaches to predict these interactions can significantly reduce the need for labor-intensive experiments, offering a more efficient and cost-effective

Table 2: Detailed specifications of Omics language models analyzed in the study.

| Model | Omics | Num Parameters (M) | Max Token length | Pre-trained Data | Tokenizer | Positional Embedding |
|---|---|---|---|---|---|---|
| DNABERT2 | DNA | 114.79 | 128 | Multispecies DNA | BPE | ALiBi |
| NTv2 | DNA | 94.00 | 1,000 | Multispecies DNA | Non-overlap 6mer | RoPE |
| RNA-FM | RNA | 99.90 | 1,024 | Multispecies ncRNA | Single | APE |
| BEACON-B | RNA | 86.12 | 1,024 | Human ncRNA | Single | ALiBi |
| ESM-1b | Protein | 653.11 | 1,024 | Multispecies Protein | Single | APE |
| ESM-2 | Protein | 149.17 | 1,024 | Multispecies Protein | Single | RoPE |
| LucaOne | Multi-omics | 1,596.31 | 1,280 | Multispecies DNA-RNA-Protein | Single | RoPE |
| CaLM | CDS | 85.70 | 1,024 | Codon Sequence | Non-overlap 3-mer | RoPE |

solution. This task is essential for understanding RNA-protein regulatory mechanisms and their roles in diverse biological processes.

**CRISPR Off-Target Prediction (CRI-Off)** involves predicting the likelihood and frequency of off-target effects by inputting both the sgRNA sequence and the corresponding off-target DNA sequences. The specificity of sgRNA is measured by a continuous target variable $y \in R$, reflecting the frequency of off-target cleavage events. The dataset from DeepCRISPR (Chuai et al., 2018) used for evaluation includes approximately 160,000 potential off-target sites from 30 sgRNAs across various cell types. SCC is employed as the performance metric to assess the relationship between predicted and observed off-target effects.

*Impact:* Precision in off-target predictions is crucial for advancing CRISPR technology, as it minimizes unintended genetic modifications that could result in harmful effects. Accurate off-target analysis is essential for refining sgRNA designs, thereby improving both the safety and efficacy of CRISPR applications in clinical and research settings. By ensuring that sgRNAs specifically target the desired DNA sequences, scientists can reduce the risk of unwanted mutations, making CRISPR-based therapies and genetic modifications more reliable and effective.

**DNA-Protein Folding Prediction (DPF)** predict the 3D structure of DNA-protein complexes by inputting their respective sequences. The training data is sourced from experimentally determined structural files found in DeepPBS (Mitra et al., 2024), which provide verified DNA-protein interaction data. From these datasets, pairs of DNA and protein sequences are extracted, along with their spatial coordinates. These coordinates serve as the labels for model training, allowing the model to learn the spatial relationships in the DNA-protein complex. The local Distance Difference Test (LDDT) is used as the evaluation metric to assess the accuracy of the predicted structures compared to the experimentally determined ones.

*Impact:* Transcription factors are essential for regulating various biological processes, including gene expression and cellular functions (Spitz & Furlong, 2012). Predicting protein-DNA binding specificity is crucial for understanding gene regulation, as proteins interact with DNA target sites with varying degrees of specificity. However, predicting binding specificity across different protein families remains a significant challenge (Chiu et al., 2023). Artificial intelligence can help overcome this by utilizing structural information from protein-DNA complexes to generalize predictions, enabling more accurate forecasts across protein families.

## 4 MODELS

We consider two types of baseline models in our benchmarks, including naive supervised models and pre-trained omics language models. We list the details in the following part in Table 2.

**Naive Supervised Models.** We employ three widely-used sequence encoders: CNN, ResNet and LSTM following the setting of BEACON and TAPE.

**Pre-trained Omics Language Models.** We evaluated the performance of single-omics, multi-omics and cross-omics language models. For single-omics language models, we utilize DNABERT2 (Zhou et al., 2023), NTv2 (Dalla-Torre et al., 2023), RNA-FM (Chen et al., 2022), BEACON-B (Ren et al., 2024), ESM-1b (Rives et al., 2021) and ESM-2 (Lin et al., 2023) for DNA, RNA, protein, cross-molecule and multi-molecule tasks. These models vary significantly in size, ranging from 86M to 653M parameters, and are pre-trained on diverse data sources including multispecies DNA, multispecies ncRNA, human ncRNA and multispecies protein. For multi-omics language models, we employ LucaOne (He et al., 2024), which is composed of a total of 1.8 billion parameters and is pre-trained on Multispecies DNA-RNA-Protein data. For cross-omics language models, Calm (Outeiral & Deane, 2024) incorporates codon embeddings into the protein language model, thereby utilizing codon sequences as pre-trained data.

Table 3: Results of different models on single-molecular tasks.

| Model/Task | GE | EA (Dev) | EA (Hk) | APA | PRS | SSP | Cont | Ther | EC |
|---|---|---|---|---|---|---|---|---|---|
| Metric | $R^2$(%) | PCC(%) | PCC(%) | $R^2$(%) | $R^2$(%) | F1(%) | P@ L/5(%) | SCC(%) | Fmax(%) |
| Literature SOTAs | | | | | | | | | |
| Literature | Xpresso | DeepSTARR | DeepSTARR | APARENT | MLP-O | UFold | MSATrans | ESM-1v | SaProt-GearNet |
| SOTA | 44.06 | 68.00 | 74.00 | 50.82 | 55.67 | 65.40 | **82.10** | **78.00** | **88.90** |
| Naive Supervised Model | | | | | | | | | |
| CNN | 34.52 | 66.08 | 74.29 | 50.93 | 45.20 | 49.95 | 5.54 | 55.86 | 53.65 |
| ResNet | 38.65 | 67.41 | 75.84 | 56.45 | 55.33 | 57.26 | 7.69 | 54.17 | 64.09 |
| LSTM | 41.34 | **68.93** | 77.02 | 67.03 | 56.54 | 58.61 | 6.07 | 58.90 | 55.58 |
| Pretrained Omics Language Model | | | | | | | | | |
| DNABERT2 | 46.40 | 68.22 | 77.43 | **72.40** | 54.79 | 24.05 | 3.84 | 16.54 | 6.69 |
| NTv2 | **48.42** | 66.20 | 76.51 | 68.75 | 55.27 | 39.76 | 27.72 | 60.19 | 48.08 |
| RNA-FM | 40.07 | 68.87 | **77.76** | 70.32 | 55.98 | 68.50 | 5.12 | 55.31 | 28.09 |
| BEACON-B | 36.32 | 66.05 | 76.31 | 70.59 | 54.67 | 64.18 | 2.72 | 60.76 | 48.84 |
| ESM-1b | 26.37 | 62.21 | 73.81 | 68.82 | 54.42 | 57.87 | 45.08 | 70.94 | 88.48 |
| ESM-2 | 34.77 | 68.04 | 77.03 | 69.52 | 56.27 | **68.75** | 55.54 | 69.36 | 85.46 |
| LucaOne | 47.70 | 68.54 | **77.76** | 69.25 | **58.26** | 56.47 | 27.31 | 68.30 | 81.11 |
| Pretrained Omics Language Model (Frozen) | | | | | | | | | |
| DNABERT2 | 13.82 | 39.44 | 41.75 | 40.48 | 19.99 | 13.51 | 11.24 | 60.94 | 47.36 |
| NTv2 | 13.78 | 29.36 | 28.89 | 30.86 | 23.09 | 14.72 | 7.48 | 60.95 | 40.57 |
| RNA-FM | 37.15 | 36.31 | 37.86 | 32.88 | 20.05 | 64.43 | 2.07 | 56.80 | 29.93 |
| BEACON-B | 23.61 | 34.57 | 38.63 | 41.21 | 25.73 | 58.73 | 12.15 | 57.18 | 39.08 |
| ESM-1b | 28.97 | 51.71 | 63.31 | 53.11 | 52.29 | 31.25 | 39.09 | 69.83 | 88.17 |
| ESM-2 | 28.99 | 43.27 | 54.73 | 27.38 | 35.51 | 44.08 | 43.34 | 63.02 | 77.80 |
| LucaOne | 41.48 | 46.86 | 51.44 | 40.11 | 38.95 | 56.86 | 3.84 | 66.33 | 73.65 |

# 5 RESULTS

## 5.1 TRAINING SETUPS

To facilitate a rigorous comparison, we conduct comprehensive fine-tuning on all BERT-based pre-trained language models, including DNABERT-2, NTv2, RNA-FM, BEACON-B, ESM-1b, ESM-2, LucaOne, and CaLM. While all models are fine-tuned using identical training hyperparameters, LucaOne is subjected to LoRA fine-tuning, whereas full-parameter fine-tuning is applied to the remaining models. For the simpler supervised models (CNN, ResNet, and LSTM), we initialize training from scratch with analogous training configurations. We use and search the learning rate from $1 \times 10^{-6}$ to $5 \times 10^{-3}$ and keep its batch size to 32. All experiments are conducted on NVIDIA A100 GPUs. Additional information is available in Appendix A.2

## 5.2 SINGLE-MOLECULAR BENCHMARK RESULTS

In Table 3 , we report the benchmark results on single-molecular tasks, including literature SO-TAs, naive supervised models and existing omics language models. Literature SOTAs include Xpresso (Agarwal & Shendure, 2020), DeepSTARR (de Almeida et al., 2022), APARENT (Bog-ard et al., 2019), MLP-O (Angenent-Mari et al., 2020), UFold (Fu et al., 2022), MSATrans (Rao et al., 2021), ESM-1v (Meier et al., 2021), SaProt-GearNet (Su et al., 2023).

**Randomly initialized vocabulary embeddings show other omics knowledge learned during pre-training.** By fine-tuning with only replacing vocabulary embeddings, the pre-trained models for each omics can achieve comparable results on most other omics tasks.This indicates that the omics knowledge learned during pre-training is not only stored in word embeddings, but also has a con-siderable proportion in the encoder, and can achieve rapid knowledge transfer of different omics through embedding replacement. And it can also help to explore commonalities and connections between different omics.

**Protein models enhance predictive performance in DNA and RNA regulatory tasks.** ESM-2 achieves results comparable to DNA or RNA models on DNA EA task and RNA APA, and PRS tasks. Notably, it even surpasses the current best model RNA-FM on the challenging SSP task. This indicates that protein models can enhance the performance of DNA or RNA regulatory prediction tasks, aiding biologists in exploring regulatory networks between proteins and regulatory elements. This cross-omics adaptability underscores the potential for multi-omics models that integrate nu-cleotide, codon, and protein-specific features to achieve more comprehensive biological insights.

**DNA models demonstrate potential in protein and RNA Tasks due to DNA's role in the central dogma's origin.** The DNA model NTv2 achieves results close to protein models on the protein Ther task. Both DNABERT2 and NTv2 also perform comparably to RNA models on RNA tasks such as APA and PRS. This suggests that DNA models have the potential to learn the properties and functions of protein and RNA models because the DNA sequences used in pre-training include regions responsible for transcribing RNA and translating proteins. This cross-domain adaptability

Table 4: Results of different models on cross-molecular tasks.

| Model/Task | Beta-Lac | Flu | EC | Model/Task | Beta-Lac | Flu | EC |
|---|---|---|---|---|---|---|---|
| Metric | SCC(%) | SCC(%) | Fmax(%) | Metric | SCC(%) | SCC(%) | Fmax(%) |
| Codon Sequence (CDS) | | | | Protein Sequence | | | |
| Naive Supervised Model | | | | | | | |
| CNN | 47.66 | 65.67 | 18.22 | CNN | 79.31 | 67.01 | 54.28 |
| ResNet | 27.20 | 66.98 | 33.68 | ResNet | 82.45 | 67.66 | 60.49 |
| LSTM | 16.92 | 67.27 | 30.46 | LSTM | 78.51 | 67.82 | 54.41 |
| Pretrained Omics Language Model | | | | | | | |
| DNABERT2 | 60.64 | 67.40 | 30.62 | ESM-1b | 85.27 | **68.12** | 87.12 |
| NTv2 | 63.34 | 67.52 | 37.12 | ESM-2 | **89.10** | 68.09 | 85.30 |
| RNA-FM | 26.45 | 59.47 | 36.37 | LucaOne | 81.08 | 67.73 | 78.32 |
| BEACON-B | 60.02 | 67.43 | 41.74 | | | | |
| CaLM | **86.56** | 67.53 | 48.91 | | | | |
| LucaOne | 82.25 | **67.95** | 52.91 | | | | |
| Pretrained Omics Language Model (Frozen) | | | | | | | |
| DNABERT2 | 25.51 | 31.14 | 28.51 | ESM-1b | 59.96 | 52.20 | **87.32** |
| NTv2 | 42.31 | 42.10 | 20.90 | ESM-2 | 47.97 | 47.53 | 76.04 |
| RNA-FM | 9.33 | 27.58 | 11.97 | LucaOne | 29.97 | 30.70 | 20.19 |
| BEACON-B | 14.40 | 22.69 | 36.54 | | | | |
| CaLM | 43.11 | 39.73 | **73.42** | | | | |
| LucaOne | 49.14 | 50.03 | 34.06 | | | | |

showcases the inherent connection between DNA, RNA, and protein representations and emphasizes the importance of designing models that leverage this central dogma framework.

**Single-omics models achieve competitive performance in their respective tasks, especially tasks about structure.** Protein models consistently secured the best performance across all protein tasks. Similarly, DNA and RNA models achieve either the top performance or perform comparably to the best models within their respective domains. Notably, in more challenging structural prediction tasks, such as RNA SSP and Protein Cont, RNA and protein models outperformed most other models by a considerable margin. This is indicative of their robust capability in capturing intricate molecular features, making them well-suited for complex structural prediction tasks. The tailored pretraining on omics-specific data plays a significant role in this success.

**Language models outperform naive supervised models.** Language models outperform the naive supervised models on three types of single-molecular tasks. This suggests that pre-training on large-scale unlabeled data does help to uncover the latent knowledge in biological data.

### 5.3 CROSS-MOLECULAR BENCHMARK RESULTS

In Table 4, we compare the performance of existing protein models and DNA/RNA/CDS models on amino acid sequences and their corresponding codon sequences (CDS).

**CDS model demonstrates competitive performance on codon sequence data.** Experimental comparisons indicate that protein-based models applied to amino acid sequences outperform CaLM applied to corresponding codon sequences. This superior performance may be attributed to ESM models being pre-trained on extensive unlabeled protein data, which include numerous mutated sequences. Consequently, the amount of pre-training data for ESM models is significantly larger than that for CaLM, which was pre-trained on codon sequences. But we observe that on Beta-Lac and Flu tasks, codon sequence-based methods achieved comparative performance, indicating potential in downstream tasks related to codon-specific information. This finding highlights the importance of considering codon-level biases and genomic context for downstream biological applications.

**Nucleotide models have the potential to compare with the CDS model.** DNA models that were not directly pre-trained on codon sequences can still achieve results close to CaLM on tasks like Flu. Moreover, the RNA model BEACON-B can achieve results on three tasks that are close to or even surpass those of DNA models. This demonstrates that nucleotide models have the potential to implicitly learn codon patterns, enabling them to capture codon-to-amino acid mappings even when not explicitly trained for tasks. This insight underlines the adaptability of nucleotide-based models and their potential to handle a broader range of biological tasks with cross-molecular relevance.

### 5.4 MULTI-MOLECULAR BENCHMARK RESULTS

In Table 5 and Table 6, we explore the combination of existing single-omics models and the multi-omics model and other models for the multi-molecular tasks. Literature SOTAs include EPI-DLMH (Min et al., 2021), DeepAAI (Zhang et al., 2022), ncRPI-LGAT (Han & Zhang, 2023) and DeepCRISPR (Chuai et al., 2018).

Table 5: Results of different models on homo-omics multi-molecules.

| Model/Task | EPI | Model/Task | AAN | Model/Task | siRNA |
|---|---|---|---|---|---|
| Metric | MCC (%) | Metric | MCC (%) | Metric | Mixed Score (%) |
| Literature SOTA | | | | | |
| EPI-DLMH | 53.59 | DeepAAI | **54.90** | - | - |
| Naive Supervised Model | | | | | |
| CNN | 25.04 | CNN | 39.08 | CNN | 56.41 |
| ResNet | 56.76 | ResNet | 45.79 | ResNet | 61.74 |
| LSTM | 58.47 | LSTM | 39.73 | LSTM | 48.69 |
| Pretrained Omics Language Model | | | | | |
| DNABERT2+NTv2 | 57.68 | ESM-1b+ESM-2 | 49.48 | BEACON-B+RNA-FM | 49.68 |
| DNABERT2+DNABERT2 | 24.59 | ESM-1b+ESM-1b | 49.63 | BEACON-B+BEACON-B | 49.60 |
| NTv2+DNABERT2 | 12.94 | ESM-2+ESM-1b | 49.36 | RNA-FM+BEACON-B | 49.65 |
| NTv2+NTv2 | 23.54 | ESM-2+ESM-2 | 49.83 | RNA-FM+RNA-FM | 49.21 |
| LucaOne | **61.29** | LucaOne | 47.49 | LucaOne | **62.33** |
| Pretrained Omics Language Model (Frozen) | | | | | |
| DNABERT2+NTv2 | 11.67 | ESM-1b+ESM-2 | 44.31 | BEACON-B+RNA-FM | 49.86 |
| DNABERT2+DNABERT2 | 10.60 | ESM-1b+ESM-1b | 48.09 | BEACON-B+BEACON-B | 49.73 |
| NTv2+DNABERT2 | 6.47 | ESM-2+ESM-1b | 42.80 | RNA-FM+BEACON-B | 50.05 |
| NTv2+NTv2 | 13.06 | ESM-2+ESM-2 | 39.42 | RNA-FM+RNA-FM | 49.48 |
| LucaOne | 15.16 | LucaOne | 25.55 | LucaOne | 50.13 |

Table 6: Results of different models on heter-omics multi-molecules.

| Model/Task | RPI | Model/Task | CRI-Off | Model/Task | DPF |
|---|---|---|---|---|---|
| Metric | MCC (%) | Metric | SC (%) | Metric | LDDT (%) |
| Literature SOTA | | | | | |
| ncRPI-LGAT | **93.20** | DeepCRISPR | **12.60** | - | - |
| Naive Supervised Model | | | | | |
| CNN | 86.25 | CNN | 10.86 | CNN | 34.64 |
| ResNet | 87.39 | ResNet | 8.90 | ResNet | 33.14 |
| LSTM | 87.83 | LSTM | 7.63 | LSTM | 32.09 |
| Pretrained Omics Language Model | | | | | |
| ESM-1b+RNA-FM | 87.41 | RNA-FM+NTv2 | 11.74 | NTv2+ESM-1b | 41.14 |
| ESM-2+RNA-FM | 88.8 | RNA-FM+DNABERT2 | 7.36 | DNABERT2+ESM-1b | 43.54 |
| ESM-1b+BEACON-B | 88.31 | BEACON-B+NTv2 | 9.21 | NTv2+ESM-2 | 43.54 |
| ESM-2+BEACON-B | 87.99 | BEACON-B+DNABERT2 | 5.61 | DNABERT2+ESM-2 | **46.35** |
| LucaOne | 88.94 | LucaOne | 11.69 | LucaOne | 39.76 |
| Pretrained Omics Language Model (Frozen) | | | | | |
| ESM-1b+RNA-FM | 83.96 | RNA-FM+NTv2 | 5.89 | NTv2+ESM-1b | 40.59 |
| ESM-2+RNA-FM | 82.83 | RNA-FM+DNABERT2 | 3.87 | DNABERT2+ESM-1b | 39.90 |
| ESM-1b+BEACON-B | 85.64 | BEACON-B+NTv2 | 4.70 | NTv2+ESM-2 | 42.53 |
| ESM-2+BEACON-B | 84.01 | BEACON-B+DNABERT2 | 3.14 | DNABERT2+ESM-2 | 43.39 |
| LucaOne | 77.90 | LucaOne | 8.42 | LucaOne | 32.65 |

**Multi-omics model can perform better than single-molecular models.** Without freezing the backbone, LucaOne performs exceptionally well on many tasks such as siRNA, EPI, and RPI. It surpasses both the combination of the two single-omics models and the naive supervised model, highlighting the effectiveness of multi-omics models trained on integrated datasets. This demonstrates that a unified multi-omics representation can capture cross-omics dependencies better than combining task-specific single-omics models, particularly when the backbone remains trainable.

**Multi-molecular tasks still present significant challenges.** In the AAN, RPI, and CRI-Off tasks, neither the approach of combining two single-omics models nor using the multi-omics model LucaOne outperforms SOTA methods. This indicates that while multi-omics models like LucaOne show promise, they struggle in tasks requiring highly specialized architectures or domain knowledge, suggesting the need for further architectural innovations and task-specific adaptations.

## 6 CONCLUSION

In this work, we present COMET, the first comprehensive multi-omics benchmark, which encompasses 17 diverse tasks spanning DNA, RNA, Protein, cross-molecule and multi-molecule study. COMET aims to address the critical gap in standardized evaluation for kinds of omics models in biology. We explore the connections between models from different omics across various tasks, gaining insights into tasks in one omic can benefit from models trained on another. We also find that multi-omics tasks still present certain challenges. These discoveries will inform the design of future biological language models, promoting interaction and understanding among different omics rather than studying each omic in isolation. However, the current benchmark involves relatively limited models and tasks, and some downstream tasks that require additional inputs are not yet aligned. In the future, we plan to further expand the models and tasks covered, closely follow developments in cross-omics and multi-omics research, and explore the potential connections between different omics data more thoroughly.

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

# A  APPENDIX

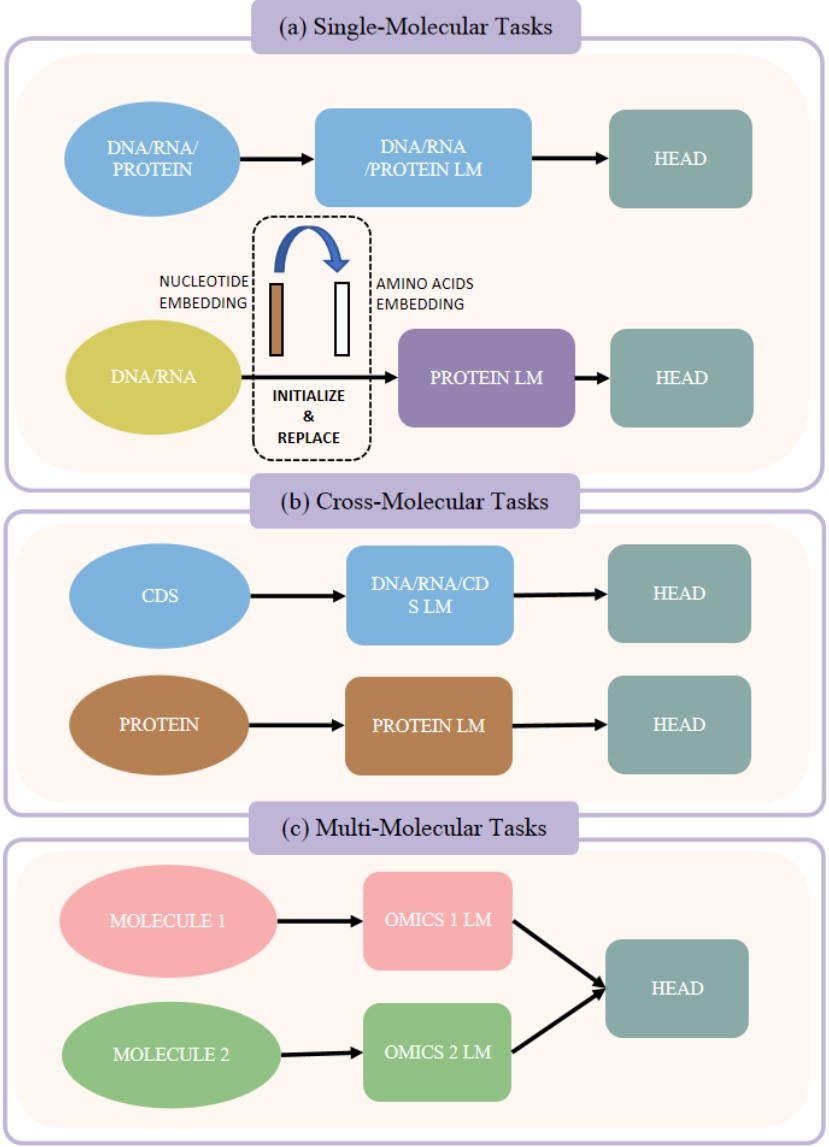

Figure 2: Detailed omics task pipeline of three kinds of tasks.

## A.1  OMICS TASK PIPELINE

These are detailed pipelines conducted in experiments and shown in Figure 2.

**Single-Molecular Task** In single-molecule tasks, in addition to testing with models and tasks belonging to the same molecule, we also evaluate the performance of models and tasks across different molecules. To address the issue of inconsistent vocabularies, we reinitialized the vocab embeddings while loading the pretrained weights. For example, DNA/RNA models initialize and replace the nucleotide vocab embedding with initial amino acids vocab embedding when using protein data.

**Cross-Molecular Task** Cross-molecule tasks involve the corresponding protein sequences and codon sequences. Therefore, we first use open-source tools and datasets to collect and construct the CDS-Protein data. Subsequently, we can input CDS data for DNA/RNA/CDS models and input

protein data for protein models. Since Lucaone is a multi-omics model capable of processing both CDS and protein sequences, these inputs are handled separately.

**Multi-Molecular Task** In multi-molecule tasks, we use different omics models with different omics molecules. By rotating foundation models within single-omics, we achieved a unified evaluation of multi-omics data. For example, a DNA-DNA task can be processed by any combination of two models within the DNA foundation model.

## A.2    BIOLOGY TASK PIPELINE

**Sequence Level Prediction** The architecture for sequence-level prediction tasks varies depending on whether the task involves single-molecule or multi-molecule scenarios.

For single-molecule sequence-level predictions, we utilize different strategies based on the model type. For naive supervised models, we compute an attentive weighted sum of all nucleotides to form a single sequence representation, which is then passed through an MLP to produce the final predictions. In contrast, when using language models, the [CLS] token representation is extracted and fed into a classifier layer to obtain the output predictions.

For multi-molecule sequence-level prediction tasks, the two interacting molecular sequences are processed independently using the same approaches as in the single-molecule scenario. Specifically, each molecule's sequence is encoded separately using either a naive supervised model or a language model to generate individual representations. The resulting embeddings are then concatenated. In specific cases, such as sgRNA and CRI-Off, additional data features are incorporated along with the concatenated embeddings. This combined representation is passed through an MLP layer to generate the final sequence-level predictions.

**Nucleotide Level Prediction** To investigate the relationships between nucleotides, we calculate the self-outer product of the nucleotide representations, resulting in a matrix that captures the pairwise interactions among nucleotides. This interaction matrix is then processed through a simple ResNet architecture to produce the final output.

**Amino Acids Level Prediction** To study the relationships between residue pairs, we employed two approaches. For the transformer-based foundational language model, we extracted the attention matrix from the backbone encoder and appended a linear layer to directly predict the residue pair relationships. For the naive supervised model, we computed the feature matrix by calculating the point-wise product between token pairs, followed by a linear layer for relationship prediction.

### A.2.1    MAX SEQUENCE LENGTH

Table 7 presents the maximum nucleotide and protein sequence lengths for tokenizers of each language model. Models such as DNABERT2, NTv2, and CaLM utilize relative positional encoding, providing excellent scalability to handle long sequences. Additionally, the BPE, non-overlap 6-mer, and non-overlap 3-mer tokenizer efficiently reduce the number of tokens, enabling these models to accommodate longer sequences under memory constraints. As a result, the maximum sequence lengths for DNABERT2, NTv2, and CaLM are set to 6000, 6002, and 3000, respectively. If a nucleotide or protein sequence exceeds the maximum length during tokenization, the tokenizer truncates the sequence to the specified limit, preserving the leftmost portion of the sequence.

Table 7: Max sequence length for tokenizers

| Models | Max sequence length |
|---|---|
| DNABERT2 | 6000 |
| NTv2 | 6002 |
| RNA-FM | 1024 |
| BEACON-B | 1024 |
| ESM-1b | 1024 |
| ESM-2 | 1024 |
| LucaOne | 1280 |
| CaLM | 1024 |

## A.3 EXPERIMENTAL SETTINGS FOR TASKS

### A.3.1 DNA TASKS

DNA tasks, including Gene expression and enhancer activity prediction are trained using the settings shown in Table 8.

Table 8: Configuration settings for DNA tasks

| Config/Task | GE | EA |
|---|---|---|
| optimizer | AdamW | AdamW |
| optimizer epsilon | 1.00E-08 | 1.00E-08 |
| optimizer momentum | $\beta_1, \beta_2 = 0.9, 0.999$ | $\beta_1, \beta_2 = 0.9, 0.999$ |
| weight decay | 0.01 | 0.01 |
| learning rate sch. | linear decay | linear decay |
| learning rate | [1e-5,5e-3] | [1e-5,5e-3] |
| warmup steps | 100 | 50 |
| epochs | 25 | 30 |
| total batch size | 32 | 32 |
| dtype | float16 | float16 |

### A.3.2 RNA TASKS

APA isoform prediction and programmable RNA switches are trained using the settings shown in Table 9. And secondary structure prediction is trained using the settings shown in Table 10.

Table 9: Configuration settings for APA isoform prediction and programmable RNA switches

| Config/Task | APA | PRS |
|---|---|---|
| optimizer | AdamW | AdamW |
| optimizer epsilon | 1.00E-08 | 1.00E-08 |
| optimizer momentum | $\beta_1, \beta_2 = 0.9, 0.999$ | $\beta_1, \beta_2 = 0.9, 0.999$ |
| weight decay | 0.01 | 0.01 |
| learning rate sch. | linear decay | linear decay |
| learning rate | [1e-5,5e-3] | [1e-5,5e-3] |
| warmup steps | 50 | 50 |
| epochs | 30 | 30 |
| total batch size | 32 | 32 |
| dtype | float16 | float16 |

Table 10: Configuration settings for secondary structure prediction

| Config/Task | SSP |
|---|---|
| optimizer | Adam |
| optimizer epsilon | 1e-08 |
| optimizer momentum | $\beta_1, \beta_2 = 0.9, 0.999$ |
| learning rate sch. | cosine decay |
| learning rate | [1e-5,5e-3] |
| warmup epochs | 1 |
| epochs | 100 |
| total batch size | 32 |
| dtype | float16 |

### A.3.3 PROTEIN TASKS

All protein tasks, including thermostability prediction, enzyme commission number prediction and contact map prediction are trained using the settings shown in Table 11.

Table 11: Configuration settings for protein tasks

| Config/Task | Cont | Ther | EC |
|---|---|---|---|
| optimizer | AdamW | AdamW | AdamW |
| optimizer epsilon | 1.00E-08 | 1.00E-08 | 1.00E-08 |
| optimizer momentum | $\beta_1, \beta_2 = 0.9, 0.98$ | $\beta_1, \beta_2 = 0.9, 0.98$ | $\beta_1, \beta_2 = 0.9, 0.98$ |
| weight decay | 0.01 | 0.01 | 0.01 |
| learning rate sch. | constant | constant | constant |
| learning rate | [2e-5,1e-3] | [2e-5,1e-3] | [2e-5,1e-3] |
| epochs | 50 | 200 | 100 |
| total batch size | 8 | 8 | 16 |
| dtype | float16 | float16 | float16 |

### A.3.4 CROSS-MOLECULER CDS/PROTEIN TASKS

In cross-molecule tasks, when the inputs are codon sequences, the settings used are shown in Table 12, while when the inputs are protein sequences, the settings used are shown in Table 13.

Table 12: Configuration settings for cross-molecules codon sequence inputs

| Config/Task | Beta-Lac | Flu | EC |
|---|---|---|---|
| optimizer | AdamW | AdamW | AdamW |
| optimizer epsilon | 1.00E-08 | 1.00E-08 | 1.00E-08 |
| optimizer momentum | $\beta_1, \beta_2 = 0.9, 0.999$ | $\beta_1, \beta_2 = 0.9, 0.999$ | $\beta_1, \beta_2 = 0.9, 0.999$ |
| weight decay | 0.01 | 0.01 | 0.01 |
| learning rate sch. | linear decay | linear decay | linear decay |
| learning rate | [1e-5,5e-3] | [1e-5,5e-3] | [1e-5,5e-3] |
| warmup steps/epoch | 100 | 100 | 50 |
| epochs | 100 | 100 | 100 |
| total batch size | 32 | 32 | 32 |
| dtype | float16 | float16 | float16 |

Table 13: Configuration settings for cross-molecules protein sequence inputs

| Config/Task | Beta-Lac | Flu | EC |
|---|---|---|---|
| optimizer | AdamW | AdamW | AdamW |
| optimizer epsilon | 1.00E-08 | 1.00E-08 | 1.00E-08 |
| optimizer momentum | $\beta_1, \beta_2 = 0.9, 0.98$ | $\beta_1, \beta_2 = 0.9, 0.98$ | $\beta_1, \beta_2 = 0.9, 0.98$ |
| weight decay | 0.01 | 0.01 | 0.01 |
| learning rate sch. | constant | constant | constant |
| learning rate | [2e-5, 1e-3] | [2e-5, 1e-3] | [2e-5, 1e-3] |
| warmup steps/epoch | 0 | 0 | 0 |
| epochs | 100 | 100 | 200 |
| total batch size | 64 | 64 | 16 |
| dtype | float16 | float16 | float16 |

### A.3.5 MULTI-MOLECULER TASKS

EPI, siRNA and AAN tasks are trained using the settings shown in Table 14. And RPI, CRI-Off tasks are trained using the settings shown in Table 15. The training settings of DPF are shown in Table 16.

Table 14: Configuration settings for EPI, siRNA and AAN

| Config/Task | EPI | siRNA | AAN |
|---|---|---|---|
| optimizer | AdamW | AdamW | AdamW |
| optimizer epsilon | 1.00E-08 | 1.00E-08 | 1.00E-08 |
| optimizer momentum | $\beta_1, \beta_2 = 0.9, 0.999$ | $\beta_1, \beta_2 = 0.9, 0.999$ | $\beta_1, \beta_2 = 0.9, 0.999$ |
| weight decay | 0.01 | 0.01 | 0.01 |
| learning rate sch. | constant | constant | constant |
| learning rate | [5e-6, 5e-5] | [5e-6, 5e-5] | [5e-6, 5e-5] |
| warmup steps | 50 | 50 | 50 |
| epochs | 30 | 30 | 30 |
| total batch size | 64 | 64 | 32 |
| dtype | float16 | float16 | float16 |

Table 15: Configuration settings for RPI and CRI-Off

| Config/Task | RPI | CRI-Off |
|---|---|---|
| optimizer | AdamW | AdamW |
| optimizer epsilon | 1.00E-08 | 1.00E-08 |
| optimizer momentum | $\beta_1, \beta_2 = 0.9, 0.999$ | $\beta_1, \beta_2 = 0.9, 0.999$ |
| weight decay | 0.01 | 0.01 |
| learning rate sch. | constant | constant |
| learning rate | [5e-6, 5e-5] | [5e-6, 5e-5] |
| warmup steps/epoch | 50 | 50 |
| epochs | 30 | 30 |
| total batch size | 64 | 32 |
| dtype | float16 | float16 |

Table 16: Configuration settings for Dna-Protein Folding

| Config/Task | DPF |
|---|---|
| optimizer | AdamW |
| optimizer epsilon | 1.00E-08 |
| optimizer momentum | $\beta_1, \beta_2 = 0.9, 0.999$ |
| weight decay | 0.05 |
| learning rate sch. | cosine decay |
| learning rate for DNA or protein model | [0,3e-5] |
| learing rate for ResNet and diffusion model | [0,1e-4] |
| warmup ratio | 10% |
| epochs | 100 |
| batch size for DNA or protein model | 1 |
| batch size for ResNet and diffusion modle | 1 |
| dtype | float16 |

## A.4 Detailed Data Preprocessing For each Task

### A.4.1 Gene Expression

We adopt the data processing methodology from Xpresso (Agarwal & Shendure, 2020). Human gene expression data comes from the Epigenomics Roadmap Consortium, which provides normalized RNA-seq values for protein-coding mRNAs across 56 tissues and cell lines.

Due to the large number of parameters in biological language models and the memory limitations of A100 GPUs, our experiments show that trimming sequence lengths to 6000 bp ensures compatibility with all models for processing input sequences. By inputting consecutive 6000 bp nucleotide fragments from different positions in the processed sequences into the Xpresso model, we identify that the sequence indexed from position 7000 to 12999 (length 6000 bp) achieves optimal test performance. This segment contains the most information related to gene expression levels.

For training, we use the 6000 bp nucleotide sequence indexed from position 7000 to 12999 as input and the expression data for 56 tissues as labels. The train, validation, and test dataset splits follow the methodology used in Xpresso.

### A.4.2 Enhancer Activity Prediction

We follow the processing procedure described in (de Almeida et al., 2022). The data includes sequence information and transcriptional activity metrics for both Drosophila and humans, encompassing developmental and housekeeping transcriptional activity levels.

We use downloaded sequences of 249 bp in length, along with `Dev_log2_enrichment_scaled` and `Hk_log2_enrichment_scaled`, which respectively represent developmental and housekeeping transcriptional activity information. The dataset is divided into training, validation, and test sets according to the method outlined in (de Almeida et al., 2022).

### A.4.3 APA Isoform Prediction

The preparation for IPA isoform analysis begins by filtering raw sequencing reads from all MPRAs (Shigaki et al., 2019) to retain only high-quality, full-length RNA sequences. These reads are grouped based on the randomized regions located upstream of the proximal polyadenylation site (pPAS), forming a dictionary of sequence variants for each library. To expand this dictionary, sequencing is also performed on the plasmid library, capturing members that lack expression of a distal isoform. RNA reads are then matched to dictionary entries by identifying the upstream region with the shortest Hamming distance.

Polyadenylation cleavage sites are determined for each mapped read by detecting the presence of a Poly-A tail. The cleavage positions are recorded as vectors associated with individual sequence variants, including a specific position for reads mapping to non-random distal sites. The dataset generated from this process consists of a dictionary of distinct sequence variants paired with vectors of cleavage position counts. A final filtering step ensures data quality by discarding sequences supported by fewer than 10–20 unique UMI RNA reads or those containing over 75% A-nucleotides within a 12–20 bp region, which could indicate internal priming artifacts.

We process data from 12 random 3' UTR libraries. 9 among the 12 libraries are used for training and 3 held out (the 3 held-out libraries were excluded from the current analysis). To construct a balanced test set, sequences from each library are first shuffled independently according to their read counts. These shuffled sequences are then merged using a round-robin approach, selecting one sequence from each library at a time in descending order of read count. This strategy ensures that the test set contains an even representation of high-read count sequences across all libraries. The remaining sequences are appended to the beginning of the combined library, and the training set is further shuffled to enhance randomness.For benchmarking purposes, the top 10% of high-read count sequences are prioritized. Among these, the most abundantly expressed sequences are selected for testing, ensuring a high-quality, balanced dataset for training, validation, and evaluation.

### A.4.4 PROGRAMMABLE RNA SWITCHES

We adopt the data generation pipeline described in (Angenent-Mari et al., 2020). A toehold-switch library comprising 244,000 potential trigger sequences is designed and synthesized, covering the complete genomes of 23 pathogenic viruses, the entire coding regions of 906 human transcription factors, and approximately 10,000 random sequences. Using this synthesized oligo pool, two construct libraries are created to represent the ON and OFF states, and both are transformed into BL21 E. coli. The OFF library includes toehold-switch constructs without triggers, while the ON library contains identical toeholds paired with complementary triggers fused to their respective switches.

The libraries are sorted into four bins using fluorescence-activated cell sorting (FACS), and the variants in each bin are quantified through next-generation sequencing (NGS) to determine their fluorescence distributions. After quality control, the toehold-switch library consists of 109,067 ON-state measurements, 163,967 OFF-state measurements, and 91,534 ON/OFF paired ratios, where both states are characterized for each switch. ON and OFF data are normalized to a scale of 0 to 1, with ON/OFF ratios normalized to a range of -1 to 1. Following (Angenent-Mari et al., 2020), a stringent quality control process is applied to eliminate artifacts and ensure data reliability. The quality control (QC) framework includes five levels: QC1, QC2, QC3, QC4 and QC5, where QC1 represents the lowest quality and QC5 the highest. Datasets above QC2 are utilized for training, while QC5 is reserved for testing.

### A.4.5 SECONDARY STRUCTURE PREDICTION

We follow the preprocessing steps outlined in the bpRNA-1m dataset (Danaee et al., 2018).To reduce sequence redundancy and improve dataset diversity, we implement an 80% sequence-identity threshold and cap the maximum sequence length at 500 nucleotides, following protocols described in the referenced studies. These measures are essential for minimizing overfitting and ensuring that the models are trained on a wide range of genetically diverse samples.

The dataset is divided into three subsets: a training set (TR0), a validation set (VL0), and a test set (TS0). The splitting process is randomized to eliminate potential biases and ensure an unbiased evaluation of the model's performance.

### A.4.6 PROTEIN TASKS

We obtain data of thermostability prediction, enzyme commission number prediction and contact map prediction from Saprot (Su et al., 2023). Following the guidance on github, we download data and place in the LMDB folder for supervised fine-tuning.

### A.4.7 CROSS-MOLECULER TASKS

For the enzyme commission number prediction task, to obtain the codon information corresponding to protein sequences, we use UniProtKB mapping function to convert UniProt IDs into European Nucleotide Archive entries. We then employ the Smith-Waterman algorithm to quickly match the corresponding codon sequences, filtering out all sequences that contained unknown nucleotides or where the number of matched nucleotides is not a multiple of three. For other cross-omics tasks, we adopt the data and settings from (Boshar et al., 2024).

### A.4.8 ENHANCER-PROMOTER INTERACTION PREDICTION

We follow the processing of (Min et al., 2021). We derive dataset from EPIANN (Mao et al., 2017), which includes six cell lines, GM12878, HeLa-S3, IMR90, K562, HUVEC and NHEK. To address the challenge of data imbalance, EPIANN enhanced the representation of positive samples by incorporating the upstream and downstream regions of enhancers. This approach expanded the dataset to include relevant genomic regions by defining extended windows of 3 kbp around enhancers and 2 kbp around promoters, ensuring a more comprehensive capture of the surrounding regulatory landscape.

### A.4.9 SIRNA EFFICIENCY PREDICTION

We get the dataset from SAIS (SAIS, 2020). We use the information of the reference sequence of the target gene, the sense sequence of the target gene, the sense sequence of modified siRNA and the remaining percentage of mRNA after the experiment named `gene_target_seq`, `siRNA_sense_seq`, `modified_siRNA_sense_seq` and `mRNA_remaining_pct` in dataset from SAIS, respectively.

### A.4.10 ANTIBODY-ANTIGEN NEUTRALIZABILITY PREDICTION

We follow (Zhang et al., 2022), which provides a minimal dataset specifically designed for this prediction task. This task is based on two datasets: CATNAP (Yoon et al., 2015), which focuses on HIV, and CoVAbDab (Raybould et al., 2021), which pertains to SARS-CoV-2.

HIV data is sourced from CATNAP in the Los Alamos HIV Database. Antibody (Ab) and antigen (Ag) sequences are extracted, curated to remove duplicates and missing values, and classified as neutralizing ($IC_{50} < 10 \, \mu g/ml$) or non-neutralizing ($IC_{50} \geq 10 \, \mu g/ml$). Seen and unseen Abs are split, ensuring no overlap between training, validation, and testing sets by excluding similar pairs (BlastP $\geq 90\%$). Training is conducted on seen Abs, with unseen Abs used for evaluation across 20 random dataset splits.

SARS-CoV-2 Data is collected from CoVAbDab and includes pairwise Ab–Ag instances across variants like Alpha, Beta, Delta, and Omicron. Five sequences per variant and 11 for Omicron are used. Omicron is treated as an unseen Ag, excluded from training but incorporated in relation graphs for transductive learning, enabling the identification of broad-spectrum Abs.

### A.4.11 RNA-PROTEIN INTERACTION PREDICTION

The dataset is sourced from NPInter2.0 , NPInter2.0_lncRNA , and RPI7317 . The sequences of ncRNAs and proteins are obtained from the NONCODE database , Gencode database , and UniProt database . The NPInter database integrates new datasets from literature and related resources, with a major focus on data published in recent years. Through a systematic PubMed search using keywords related to RNA interactions, 1270 relevant articles were identified. Verified or processed interaction data were manually extracted, while raw sequencing data were excluded. Binding sites were compared against RefSeq coding genes to remove overlaps with coding regions and cross-checked with NONCODE for ncRNA references. Valid interactions were annotated with standardized IDs (UniProt, RefSeq, NONCODE, etc.) depending on the molecule type.

Data from external resources like LncRNADisease, which curated 478 experimentally supported lncRNA interactions, were integrated and subjected to the same annotation pipeline. The combined dataset underwent redundancy elimination, aggregating overlapping interactions into single records. NPInter v2.0 thus provides a comprehensive, curated multilevel snapshot of RNA-related interactions.

### A.4.12 CRISPR OFF-TARGET PREDICTION

Following (Chuai et al., 2018), we get the off-target dataset, which comprises two different cell types contains 30 sgRNAs. For all 30 sgRNAs, approximately 160,000 possible off-target sites across the entire genome are obtained. Off-target sites are annotated and standardized using the targeting cutting frequency (indel frequency) detected by different off-target detection methods.

### A.4.13 DNA-PROTEIN FOLDING PREDICTION

We query the PDB database using the filenames provided by deepPBD (Mitra et al., 2024) to obtain the mmCIF files of DNA-protein complexes and get 428 mmCIF files. From the mmCIF files, we extract the coordinates, sequences, and certain bonding information of both DNA and proteins. When encountering modified residues or nucleotides in the mmCIF files, we follow the AlphaFold3 (Abramson et al., 2024) and map these residues or nucleotides to standard amino acids or DNA sequences using SCOP. We set the DNA-protein interface distance threshold to 5 Å. Based on this threshold, we derive the DNA-protein interface information. Subsequently, we match the

DNA and protein duplex information using the DNA-protein interface and sequence information. Finally, we obtained 683 DNA-protein complexes.

## A.5 MULTI-MOLECULER TASKS

### A.5.1 SIRNA EFFICIENCY PREDICTION

We use siRNA modification features combined with sequence features to enhance mRNA target specificity.

For the metric, we use mixed score as the metric-a custom metric balances regression error and classification accuracy by integrating F1 score (harmonic mean of precision and recall), Mean Absolute Error (MAE), and Rnage-MAE (MAE computed within a range threshold) (SAIS, 2020).

The formula for Range-MAE is:

$$Range\text{-}MAE = \frac{1}{m} \sum_{i=1}^{m} |y_i - \hat{y}_i|,$$

where $m$ is the number of samples with predicted values within the range $[0, 30]$. The Range-MAE evaluates the average absolute error of predictions in this specific range, with values in $[0, 100]$.

F1 combines precision and recall to evaluate the classification performance of predictions within the Remaining range, focusing on $[0, 30]$. It outputs a final F1 score between $[0, 1]$.

The mixed score will be calculated based on the following formula:

$$Mixed - score = 50\% \times (1 - \frac{MAE}{100}) + 50\% \times F1 \times (1 - \frac{Range\text{-}MAE}{100})$$

The first part of the score formula focuses on the overall accuracy of the model, while the second part emphasizes prediction precision within the low Remaining range.

### A.5.2 CRISPR OFF-TARGET PREDICTION

We use epigenetic features including CTcF, Dnase, H3K4me3 and RRBs combined with sequence features to enhance CRISPR target specificity.

### A.5.3 DNA-PROTEIN FOLDING

The model architecture for Dna-Protein Folding task imitates AlphaFold3. We engage DNA and protein models as input embedders to get DNA and protein representations, respectively. The DNA and protein representations are transformed to form DNA-protein single representations and DNA-protein single representations, and fed into a ResNet to get the final DNA-protein single and pair representations. The diffusion module processes the final DNA-protein single and pair representations to generate the structure of DNA-protein complexes.

Differ from AlphaFold3, we replace the template module, the MSA module and the pairformer module with a ResNet architecture composed of eight residual blocks. Furthermore, we modify the original diffusion module to consist of two encoder blocks, two decoder blocks, and four diffusion transformer blocks. The losses consist of diffusion loss and distogram loss:

$$\mathcal{L}_{\text{loss}} = \alpha_{\text{diffusion}} \cdot \mathcal{L}_{\text{diffusion}} + \alpha_{\text{distogram}} \cdot \mathcal{L}_{\text{distogram}}$$

where $\alpha_{\text{diffusion}} = 4.0$ and $\alpha_{\text{distogram}} = 0.03$. Each item in the losses is followed to AlphaFold3.

## A.6 METHODS IN BENCHMARK

All pre-trained biology foundation models utilize a Masked Language Modeling (MLM) objective.

For the MLM task, an input sequence is given, where 15% of its elements are randomly masked. The model processes this masked sequence and aims to predict the original elements. This strategy mirrors the Cloze test found in conventional language modeling.

- 15% of the tokens in the sequence are masked.

- In 80% of the cases, the masked tokens are replaced by a special MASK token.

- In 10% of the cases, the masked tokens are substituted with a random token different from the original.

- In the remaining 10% of cases, the masked tokens remain unchanged.

### A.6.1 DNABERT2

**Training Data** DNABERT2 utilized datasets from the human genome and multiple species genomes, totaling 35.24 billion nucleotide bases. The human genome dataset comprised 2.75 billion nucleotide bases, while the multi-species genome dataset included 32.49 billion nucleotide bases from the genomes of 135 different species. During the data processing, all sequences containing 'N' were removed, leaving only those composed of ATCG nucleotides.

### A.6.2 NTv2

**Training Data** NTv2 leverages three datasets: the Human reference genome dataset, which contains 3.2 billion nucleotides; the 1000 Genomes Project (1000G) dataset, featuring over 20.5 trillion nucleotides; and the Multispecies dataset, comprising 174 billion nucleotides from 850 species.

During the preprocessing phase, all nucleotides outside of ATCG are replaced with 'N'. For both the multispecies and human reference datasets, the genomes are segmented into overlapping chunks of 6,100 nucleotides. Each chunk overlaps with the previous one by sharing the first 50 nucleotides and with the next one by sharing the last 50 nucleotides.

### A.6.3 RNA-FM

**Training Data** The RNA-FM model is pre-training utilizing data sourced from RNACentral. To ensure the non-redundancy of the dataset, RNA-FM employs CD-HIT (specifically, CD-HIT-EST) with a threshold set at 100% sequence identity. This process led to a final dataset comprising 23.7 million distinct RNA sequences.

### A.6.4 BEACON-B

**Training Data** The BEACON-B model uses 523,934 human ncRNA sequences filtered from the total ncRNA in the RNACentral database Sweeney et al. (2020) as pre-training data.

### A.6.5 ESM1B

**Training Data** The ESM-1b model was pre-trained on Uniref50, which comprises approximately 30 million protein sequences. During the training process, sequences exceeding 1023 tokens (excluding the CLS token) are randomly truncated to a length of 1023 tokens.

### A.6.6 ESM2

**Training Data** The ESM-2 model is trained using the UniRef50 dataset. To enhance the data volume and diversity, during each training update, a mini-batch of sequences from UniRef50 is sampled and replaced with sequences uniformly sampled from the corresponding UniRef90 clusters. This approach allows the ESM-2 model to be trained on over 60 million protein sequences.

### A.6.7 CALM

**Training Data** The CaLM model is pre-trained using cDNA data collected from the European Nucleotide Archive database. During preprocessing, sequences containing unknown nucleotides, start codons that are not ATG, internal stop codons, or a nucleotide count not divisible by three are removed. The final dataset consists of about 9 million cDNA sequences.

### A.6.8 PROS AND CONS ANALYSIS

We can see that all models are trained using MLM, among which the NTv2 model uses much more pre-training data than other models, which may make it more generalizable on more tasks. The RNA-based models are all pre-trained only on ncRNA, and BEACON-B only uses a small part of the pre-training data, which may affect their potential performance. Models like the protein-based model and the Calm codon model use amino acid (non-overlapping 3mer) encoding, which may have an advantage in tasks that focus on amino acid expression. For the RNA-FM and ESM-1b models, the use of absolute position encoding limits the length of their input sequences, which may affect their performance on long sequence tasks.

## A.7 LoRA FINE-TUNING SETTINGS

We utilized LoRA fine-tuning to optimize LucaOne. The configuration settings for LoRA fine-tuning are presented in Table 17.

Table 17: Configuration settings of LoRA fine-tuning for LucaOne

| Config | Value |
| --- | --- |
| Weight Type | $W_q, W_k, W_v, W_o$ |
| LoRA rank | $r_q = r_k = r_v = r_o = 32$ |
| LoRA $\alpha$ | 32 |
| Dropout Prob | 0.05 |

## A.8 FURTHER DISCUSSION

We appreciate the reviewer's suggestion and agree that future directions are crucial for expanding the impact of this work. We propose three potential areas for further exploration:

- Incorporating Additional Omics Data and Biological Priors: Expanding the benchmark to include additional omics data types and features that indirectly influence biological processes in downstream tasks will enhance its comprehensiveness. Integrating biological priors, such as pathway-level annotations, protein-protein interactions, or chromatin accessibility maps, can provide a more holistic view of molecular interactions and improve the relevance of downstream predictions.

- Refining the Evaluation Pipeline for Multimodal Data: Developing a more sophisticated evaluation pipeline that better infuses multimodal data will be essential. For instance, metrics that capture cross-modality consistency and assess how well models leverage complementary information from multiple omics types can provide deeper insights into model performance. Additionally, incorporating metrics that account for the stochasticity and uncertainty inherent in biological systems can improve evaluation robustness.

- Developing Novel Multi-Omics Foundation Models: Leveraging the insights gained from this benchmark, we aim to explore novel model architectures tailored for multi-omics data. These models could employ advanced techniques like attention-based integration and hierarchical representations of omics modalities. The benchmarking insights will guide the design of these models, ensuring they address the specific challenges and opportunities identified in multi-omics tasks.

These future directions aim to expand the scope and utility of the benchmark, driving the development of innovative methods and fostering deeper biological insights through more accurate and comprehensive modeling of multi-omics data.

## A.9 ASSETS

### A.9.1 SOFTWARE AND LIBRARIES

The open-source software, and corresponding licenses are presented in Table 18. The data, licenses and corresponding URL are presented in Tab. 19Table 20.

Table 18: Software used in this work

| Asset | License |
|---|---|
| FlashAttention | BSD-3-Clause |
| Pytorch | BSD-3-Clause |
| Pytorch Lightning | Apache-2.0 |
| Huggingface | Apache-2.0 |
| Scikit-Learn | BSD-3-Clause |
| Numpy | BSD-3-Clause |
| Matplotlib | Matplotlib License |
| Seaborn | Apache-2.0 |

Table 19: Dataset used in this work(Part 1)

| Dataset | Sub-dataset | License | URL |
|---|---|---|---|
| Enhancer Activity | Deepstarr | MIT | https://zenodo.org/records/5502060 |
| Gene Expression | Xpresso | MIT | https://xpresso.gs.washington.edu |
| | GTEx | | https://www.gtexportal.org/home/ |
| APA Isoform | APARENT | MIT | shttps://github.com/johli/aparent |
| Programmable RNA Switches | GEO | MIT | https://www.ncbi.nlm.nih.gov/geo/query/acc.cgi?acc=GSE149225 |
| Secondary Structure | | | https://bprna.cgrb.oregonstate.edu/about.php |
| | CRW | | https://crw-site.chemistry.gatech.edu |
| | tmRDB | Research Purpose Only | https://rth.dk/resources/rnp/tmRDB/ |
| | SRPDB | Research Purpose Only | https://rth.dk/resources/rnp/SRPDB/ |
| | tRNADB | | http://trna.bioinf.uni-leipzig.de/DataOutput/ |
| | Rnase P | Public Domain | |
| | RFam | CC0 1.0 | https://rfam.org |
| | PDB | CC0 1.0 | https://www.rcsb.org |
| Thermostability | | AFL-3.0 | https://benchmark.protein.properties/ |
| Contact Map | ProteinNet | MIT | https://github.com/aqlaboratory/proteinnet |
| Function EC | PDB | CC0 1.0 | https://www.rcsb.org |

Table 20: Dataset used in this work(Part 2)

| Dataset | Sub-dataset | License | URL |
|---|---|---|---|
| | SWISS-MODEL | CC BY-SA 4.0 Creative Commons Attribution-ShareAlike 4.0 International License | `https://swissmodel.expasy.org/` |
| Enhancer Promoter Interaction | EPIANN | | `https://github.com/wgmao/EPIANN` |
| siRNA Efficiency | Shanghai Academy of AI for Science | | `http://competition.sais.com.cn/` |
| Antibody-Antigen Neutralizability | CoVAbDab | CC-BY 4.0 license | `https://opig.stats.ox.ac.uk/webapps/covabdab/` |
| | CATNAP | Non Commerical | `shttps://www.hiv.lanl.gov/components/sequence/HIV/neutralization/download_db.comp` |
| RNA-Protein Interaction | NPInter2.0 | | `http://www.bioinfo.org/NPInter` |
| | RPI7317 | | `https://github.com/NWPU-903PR/LPI_BLS` |
| CRISPR Off Target | DeepCRISPR | Apache-2.0 license | `https://github.com/bm2-lab/DeepCRISPR` |
| Protein-CDS-Fluorescence | NCBI | | `https://www.ncbi.nlm.nih.gov/bioproject/PRJNA282342` |
| Protein-CDS-EC | ENA | | `https://www.ebi.ac.uk/ena` |
| Protein-CDS-Beta_Lactamase (Complete) | | | `https://github.com/FowlerLab/Envision2017` |
| DNA-Protein Structure | JASPAR | Creative Commons Attribution 4.0 International License. | `https://jaspar.elixir.no` |
| | HOCOMOCO | | `https://hocomoco11.autosome.org/` |

