# OpenReview forum: "COMET: Benchmark for Comprehensive Biological Multi-omics Evaluation Tasks and Language Models"
_ICLR.cc/2025/Conference — Submitted to ICLR 2025_

### Official Review · Reviewer_8ScB · 2024-10-27

**Soundness:** 3
**Presentation:** 2
**Contribution:** 3
**Rating:** 5
**Confidence:** 4

**Summary:**

This paper proposed a new benchmarking pipeline for biological sequence models.

**Strengths:**

This paper is well-structured and covers most recent foundation models for biological sequence analysis, which will attract researchers in this field.

**Weaknesses:**

Although the pipeline and experiments are interesting, I do have some questions and I believe that the authors overclaim their dataset preparation and pipeline design. Here are my questions:

1. In Figure1, the authors mention that the benchmarking resources coming from different databases. However, if we zoom in each task, only one dataset is selected into benchmarking, which is really a biased selection. For similar benchmarking analysis [1,2], most of them will cover diverse datasets for each task. This is also different from a recent publication BEND [3], because the authors here intend to benchmark information like gene expression, which is less constant comparing with tasks like gene finding. Would the authors please justify that their choices are unbiased or consider including more datasets?

2. For Table 1, the metrics seem inconsistent. For example, why the authors sometimes select SCC, while other cases are based on PCC or R2? Should all of them be used for benchmarking a regression task? Also, what is the reason of not using PCC for all cases but using SCC for certain task? I cannot understand the reason.

3. In Table 2, the authors consider DNABERT2 [4]. However, I am confused about the max token length of DNABERT2. The current version of DNABERT2 can handle DNA sequence with different length, and the token length can be increased. Could the authors use the latest version for benchmarking to make a fair and useful comparison?

4. In Table 3, what is the meaning of percentage (%) for PCC, SCC or R2? These metrics can be negative, so what is the meaning of a negative percentage?

5. Could the authors ensure that their task-specific method (like Xpresso) is really the optimal solution? Why do the authors not choose Enformer [5] or Borzoi [6] for gene expression prediction? It will be much more helpful to list the sources of task-specific method and the reason to choose them (like based on benchmarking studies, they are top-performer?)

6. I am a bit confused from the conclusion of this benchmarking analysis. Would the authors please highlight their discovery in the abstract section and thus the readers can learn from your paper? I think most of the content of abstract and conclusion are presenting experiment design, but other key information, like some tasks need better methods, should be emphasized.

7. Furthermore, I wonder if the authors can check if the validation dataset might be used in the pre-training stage of some foundation models. If so, will the benchmarking analysis for the frozen one fair?

[1] https://www.nature.com/articles/s41592-019-0690-6

[2] https://www.nature.com/articles/s41592-022-01480-9

[3] https://arxiv.org/abs/2311.12570

[4] https://github.com/MAGICS-LAB/DNABERT_2

[5] https://www.nature.com/articles/s41592-021-01252-x

[6] https://www.biorxiv.org/content/10.1101/2023.08.30.555582v1

**Questions:**

Please see my weaknesses.

---

> ### Author Response · Authors · 2024-11-23
> **Answer to Reviewer 8ScB**
>
> We sincerely thank the reviewer for the insightful feedback. Below are our responses to the points raised:
>
> ### **Weakness1: Choices of Datasets**
>
> We curated a comprehensive collection of tasks encompassing diverse molecular types and enlisted evaluations from several biology professors and PhD students. From these assessments, we selected a representative subset of tasks to establish the first multi-omics benchmark spanning DNA, RNA, and proteins.
>
> - As shown in Table 1 of our paper, the tasks were sourced from high-impact conferences, journals, and competitions, emphasizing literature with high citation counts. Many of these tasks have already been used to evaluate the performance of biological language models specific to their respective omics. For instance, Gene Expression data originates from Cell Reports, Enhancer Activity Prediction from Nature Genetics, and APA Isoform Prediction from Cell. Similarly, tasks such as Programmable RNA Switches and Secondary Structure Prediction derive from Nature Communications and Nucleic Acids Research, respectively, while others like Thermostability Prediction and Contact Map Prediction are sourced from NeurIPS and BMC Bioinformatics. The selected tasks span both single-omics and cross-omics domains and broadly encompass the aspects of structure, function, and engineering.
>
> - Notably, for DNA tasks, we included Gene Expression Prediction, which forecasts the cross-tissue expression levels of genes and transcription factors (TFs), shedding light on regulatory networks underlying cell states and tissue-specific genomic functions. Enhancer Activity Prediction, on the other hand, analyzes DNA sequences to predict enhancer activity on specific promoters, revealing how regulatory signals drive transcriptional specificity in different cell types. These tasks also vary significantly in sequence lengths—Gene Expression tasks use sequences of 6000 bp, while Enhancer Activity Prediction involves sequences of 249 bp for evaluation of model performance across varying DNA sequence lengths. For the other expression work, BEELINE [r1] and Li et al.’s [r2] focus on cell type identification and classification in single-cell RNA sequencing (scRNA-seq) datasets and provide curated lists of marker genes for specific cell types across tissues in humans and mice. However, it is mainly used in single-cell research with non-sequence data, such as scBERT [r3]. In our study, the gene expression task utilizes the Xpresso dataset from the Epigenomics Roadmap Consortium [r4], which is at the bulk level and emphasizes the regulatory effects of cell type-specific non-coding regions on gene expression, as well as inputs into the model as the sequence data.
>
> - Looking ahead, we aim to expand the benchmark by incorporating additional high-impact tasks that span multiple omics and broaden the scope of structural and functional predictions, driving innovation in bioinformatics and computational biology.
>
> [r1] Benchmarking algorithms for gene regulatory network inference from single-cell transcriptomic data. Nature Methods 2020.
>
> [r2] Benchmarking spatial and single-cell transcriptomics integration methods for transcript distribution prediction and cell type deconvolution. Nature Methods 2022.
>
> [r3] scBERT as a large-scale pretrained deep language model for cell type annotation of single-cell RNA-seq data. Nature Machine Intelligence 2022.
>
> [r4] Integrative analysis of 111 reference human epigenomes. Nature 2015.
>
> ### **Weakness2: Metrics of Regression Tasks**
>
> We maintain assessment metrics that are consistent with the original task, facilitating alignment with the performance of the original task. Like other bio-benchmark work, such as the PEER benchmark (NeurIPS 2022) [r1] also has different metrics for regression tasks such as SCC and RMSE, the BEACON benchmark (NeurIPS 2024) [r2] uses different metrics for regression tasks such as SCC, R^2, and RMSD, and similarly, the GUE benchmark (ICLR 2024) [r3] uses different metrics for classification tasks such as MCC and F1.
>
> [r1] Peer: a comprehensive and multi-task benchmark for protein sequence understanding. NeurIPS 2022.
>
> [r2] BEACON: Benchmark for Comprehensive RNA Tasks and Language Models. NeurIPS 2024.
>
> [r3] Dnabert-2: Efficient foundation model and benchmark for multi-species genome. ICLR 2024.
>
>
> ### **Weakness3: Version of DNABERT2**
>
> We used the latest version of DNABERT2 to do the experiments. Without unfair comparison, since DNABERT2 uses extrapolatable positional encoding, we automatically adapted the length of the downstream task in our experiments, for example, in the gene expression prediction task, the token length can be up to about 1500.
> What we report in Table 2 is the maximum token length of the original DNABERT2 in the downstream task, which we have updated to the pre-trained token length.

---

> ### Author Response · Authors · 2024-11-23
> **Answer to Reviewer 8ScB**
>
> ### **Weakness4: Explanation of Percentage**
> We followed the representation in BEACON and used percentage (%) to indicate that the value is scaled by a factor of one hundred. In Table 3, PCC, SCC, and R² values are multiplied by 100 and reported as percentages (%) for readability. Negative percentages indicate negative values of the original metric scaled by a factor of 100, reflecting the same interpretation as their unscaled counterparts.
> ### **Weakness5: Justification for the Task-Specific Method Selection**
> - We ensure that the task-specific methods chosen are the most relevant and effective. Although Enformer and Borzoi also target gene expression prediction, its task differs significantly from the gene expression task in our study due to the large and computationally complex cost of the task.
> - The input sequence lengths for these tasks vary substantially: Enformer operates on sequences of length 200k, whereas Xpresso’s gene expression tasks use sequences of length 6k. The 200k sequence length exceeds the processing capability of RNA and protein language models, making Enformer unsuitable for inclusion as a comparative method for gene expression prediction tasks in this study.
> - Additionally, Enformer focuses on predicting expression levels across tracks and buckets based on CAGE data, which requires a highly complex head architecture following the language model. This approach demands significantly higher computational costs for full-parameter fine-tuning.
>
> ### **Weakness6: Emphasis on Other Key Information**
> Thanks for the suggestion, we've updated some of the other key information below, highlighted in the manuscript `ABSTRACT` in `red`.
> - We observed that DNA, RNA, and protein models can be applied to tasks across different omics by leveraging initialized embeddings, with protein models demonstrating superior performance across various omics.
> - Through the evaluation of multi-omics tasks, we identified significant gaps in the capabilities of current models to address these challenges, highlighting substantial opportunities to enhance multi-omics integration and improve overall performance.
>
> ### **Weakness7: Justifications for Prevention of Information Leakage**
> - For the unsupervised pre-training, since most models are pre-trained on sequences without corresponding labels, we do not need to worry about data overlap between the pre-training and downstream tasks.
> - For downstream tasks, we followed established, peer-reviewed data processing pipelines for tasks like secondary structure prediction and APA isoform prediction (see details in Answer to Reviewer 6UP5 `Question5: Detailed Preprocessing for Each Task`). Additionally, after preprocessing, we double-check to ensure there is no overlap between the training and testing datasets.

---

> ### Comment · Reviewer_8ScB · 2024-11-24
> **Thanks for your reply, but sorry they are not satisfiable.**
>
> Thanks for your reply, which addresses some of my concerns. However, my major concerns including the lack of data diversity as well as baselines are not fully addressed and thus I intend to keep my score. It seems that the authors are not very familiar with sequence to function modelling or molecular biology, and I do not think it is reasonable to argue that you just follow others' ideas without justifying whether their option is correct or not. This is extremely important for a successful benchmarking paper. Furthermore, Enformer is already widely used in gene expression prediction task, people even finetuned it if needed (https://www.biorxiv.org/content/10.1101/2024.07.27.605449v1). I do not understand why the authors argue the reason based on computation resources. We do need comprehensive data and baselines to make a fair conclusion.

---

> ### Author Response · Authors · 2024-11-24
> **Thank you for your feedback. To address your remaining concerns**
>
> Thank you for your feedback. To address your remaining concerns, we provide the following responses:
>
> **1. Dataset Selection:**
>
> We began by identifying key tasks in molecular biology involving DNA, RNA, and proteins and selected representative datasets for each task. Our task and dataset selection process is guided by community recognition, relying on peer-reviewed sources with real-world applications, rather than blindly following others' ideas without critically evaluating their validity.
>
> **2. Diversity of Datasets:**
>
> We carefully considered dataset diversity by ensuring:
>
> - Coverage across all omics: DNA, RNA, and protein, as well as multi-omics interactions.
> - A variety of task types: including sequence-wise regression, sequence-wise classification, residue-wise regression, and residue-wise classification.
> - A wide range of data sizes: from ~700 to 41K samples across different datasets.
> A broad and reasonable range of sequence lengths: from 23 to 6k.
> - Datasets with excessively long sequences, such as the 200k-length CAGE profile prediction dataset used by Enformer, were excluded as they are unsuitable for evaluating RNA and protein language models. In the future, we plan to include multiple datasets for the same task type to further enhance diversity within individual tasks.
>
> **3. Baseline Method Selection:**
>
> - We selected baseline methods based on their demonstrated applicability and representativeness for specific tasks. For instance, in the gene expression prediction task, Xpresso serves as an appropriate baseline for sequence-wise regression tasks predicting bulk RNA-seq expression across 56 tissues and cell lines. In contrast, Enformer’s task involves predicting CAGE profiles through binned nucleotide-wise regression. Evaluating Enformer requires datasets that match its task, and using mismatched datasets would compromise fairness (please see `notes` below).
> - This is analogous to computer vision benchmarks where segmentation tasks are divided into semantic segmentation and instance segmentation—methods designed for one task are not necessarily evaluated on the other, as their models and datasets are not aligned. While we recognize Enformer’s contributions to gene expression prediction and acknowledge that its model could be adapted for this task by modifying its head, we plan to include it in future work.
>
> **4. Computational Resource Considerations:**
>
> Our mention of computational resources primarily aims to clarify why we selected Xpresso's sequence-wise regression bulk RNA-seq dataset for the gene expression task instead of Enformer’s binned nucleotide-wise regression CAGE profile dataset.
>
> We would greatly appreciate your evaluation and feedback.
>
> `notes`：
> - Model: Xpresso, Dataset: Bulk RNA-seq, Task: sequence-wise regression, Length: 6k
> - Model: Enformer, Dataset: CAGE Profile, Task: binned nucleotide-wise regression, Length: 200k

---

> > ### Comment · Reviewer_8ScB · 2024-11-25
> > **Thanks for you reply.**
> >
> > Thanks for you reply. I keep my scores as I do not see more datasets for one specific task as a requirement of data diversity for a benchmarking paper. You can focus on addressing others' opinions to increase the probability of raising the score.

---

> > > ### Author Response · Authors · 2024-11-27
> > > **Thank you for your feedback. To address your remaining concerns**
> > >
> > > Thank you for your response. We genuinely aim to address your concerns.
> > >
> > > - Our benchmark carefully considers dataset diversity and already includes multiple datasets for the same task. For example, the enhancer activity prediction task incorporates two datasets: one for housekeeping enhancers and another for developmental enhancers. This enables a more comprehensive evaluation of enhancer regulatory capabilities from different perspectives.
> > > - Additionally, we have enriched the diversity of datasets for other tasks as well. For the APA Isoform task, we collected three additional datasets on Alternative Polyadenylation from Massively Parallel Reporter Assays and conducted corresponding experimental evaluations, as detailed in the table below.
> > >
> > > |        Model/Task        | APA_testset | HSPE1 | SNHG6 | WHAMMP2 |
> > > |:------------------------:|:-----------:|:-----:|:-----:|:-------:|
> > > |          Metric          |     R^2     |  R^2  |  R^2  |   R^2   |
> > > |      Literature SOTA     |             |       |       |         |
> > > |          APARENT         |    57.68    | 33.15 |  9.55 |  20.12  |
> > > |  Naive Supervised Model  |             |       |       |         |
> > > |            CNN           |    50.93    | 10.24 |  2.91 |   4.38  |
> > > |          ResNet          |    56.45    | 13.91 |  0.77 |   6.41  |
> > > |           LSTM           |    67.03    | 35.79 |  4.22 |  19.42  |
> > > | full parameter fine-tune |             |       |       |         |
> > > |         DNABERT2         |    72.40    | 34.69 |  0.24 |  16.32  |
> > > |           NTv2           |    68.75    | 35.09 | 11.08 |  17.53  |
> > > |          RNA-FM          |    70.32    | 39.24 |  7.34 |  16.43  |
> > > |         BEACON-B         |    70.59    | 27.54 |  0.04 |   8.71  |
> > > |          ESM-1b          |    68.82    | 42.87 | 10.99 |  16.88  |
> > > |           ESM-2          |    69.52    | 41.31 | 10.68 |  20.27  |
> > > |          LucaOne         |    69.25    | 39.78 | 10.62 |  13.74  |
> > > |      freeze backbone     |             |       |       |         |
> > > |         DNABERT2         |    40.48    | 0.56  |  0.04 |   0.17  |
> > > |           NTv2           |    30.86    |  0.96 |  0.03 |   0.24  |
> > > |          RNA-FM          |    32.88    |  0.61 |  0.91 |   0.05  |
> > > |         BEACON-B         |    41.21    |  1.64 |  0.22 |   1.97  |
> > > |          ESM-1b          |    53.11    | 23.53 |  7.91 |   9.52  |
> > > |           ESM2           |    27.38    |  1.01 |  4.73 |   0.89  |
> > > |          LucaOne         |    40.11    |  4.66 |  1.04 |   1.72  |
> > >
> > > #### **1. Performance Trends Across Datasets**
> > >
> > > - **APA_testset is the Benchmark Leader**:
> > >   - Most models achieve their best $R^2$ on this dataset, likely due to its relatively simpler or more predictable biological patterns.
> > >
> > > - **HSPE1 is Moderately Predictable**:
> > >   - Performance on HSPE1 is lower than APA_testset but higher than SNHG6 and WHAMMP2.
> > >   - Pretrained models like ESM-1b show the strongest results here.
> > >
> > > - **SNHG6 and WHAMMP2 are the Most Challenging**:
> > >   - Both datasets consistently yield the lowest $R^2$ scores, with SNHG6 performing particularly poorly for most models.
> > >   - This indicates a need for more sophisticated or specialized models for these datasets.
> > >
> > > #### **2. General Trend of Models Across Datasets**
> > >
> > > - **Pretrained Models Dominate**:
> > >   - Fine-tuned pretrained models (e.g., DNABERT2, ESM variants) consistently outperform classical architectures on all datasets.
> > >   - However, the advantage of pretrained models diminishes for SNHG6 and WHAMMP2.
> > >
> > > - **Classical Models Are Dataset-Dependent**:
> > >   - While classical models (e.g., LSTM) perform reasonably well on APA_testset, they struggle on other datasets, highlighting their limited generalizability.

---

> ### Comment · Reviewer_8ScB · 2024-11-28
> **Thanks for your reply**
>
> Thanks for your work, but I still intend to keep my scores. My question is more like including more diverse datasets and justify the reasons of including such datasets. As a researcher doing benchmarking analysis, you need to prove that your benchmark analysis is comprehensive and you should have the same criteria for different tasks. For example, you have included more datasets for only two tasks, how about other tasks? Does it mean we cannot find enough datasets for certain tasks, or it means it is too early to perform a benchmarking analysis for this task given limited resources?
>
> Back to the two tasks you include with more datasets, I think there exists overlap between housekeping enhancers and developmental enhancers, at least in fly (https://www.nature.com/articles/s41467-024-52921-2) and their working principles are different. How to ensure that your selected datasets are independent and if the same enhancers have different outputs across these two conditions, how to interpret the results? There are many things we need to consider other than reporting scores.
>
> Finally, I intend to emphaize that I am not a bad reviewer, I do not perform benchmarking analysis like you did (so there is no conflict) but I am interested in using a good DNA sequence morel for exciting applications, that is why I am very serious of evaluating a benchmarking manuscript. I think you have much more space to explore and improve your manuscript to finally make a great contribution in this field.

---

> > ### Author Response · Authors · 2024-12-01
> > **Thank you for your feedback.**
> >
> > Thank you for your thoughtful feedback and for your interest in the diversity of datasets in our benchmark, as well as for clarifying any potential conflicts of interest. We deeply value your rigorous evaluation and commitment to ensuring the quality of benchmarking analyses.
> >
> > - **Included Dataset and Diversity:** Regarding the overlap between housekeeping and developmental enhancers, we intentionally included these datasets to study two distinct types of enhancer activity. While these datasets may overlap, the aim is to understand the differences in their functional outputs under varying conditions. To further enhance dataset diversity in our benchmark, we have included additional datasets covering multiple cell types. For instance, in the Enhancer-Promoter Interaction Prediction task, we incorporated interaction scenarios from six distinct cell type datasets: `GM12878`, `HOVEC`, `Hela-S3`, `IMR90`, `K562`, and `NHEK`. Additionally, in this rebuttal, we have enriched our work with extra datasets for the task Alternative Polyadenylation, broadening the scope of isoform-level analysis with `HSPE1`, `SNHG6` and `WHAMMP2` datasets. Moreover, HSPE1 focuses on protein folding, SNHG6 addresses non-coding RNA functions, and WHAMMP2 targets protein interaction and structural dynamics, showcasing the broad functional spectrum.
> >
> > - **Benchmarking Precedents:** Based on precedents in the field, having multiple datasets for the same task is not a strict requirement for a benchmark. For instance, recently accepted benchmarks such as BEACON [r1] and DART-Eval [r2] do not include multiple datasets for every task. Similarly, other well-regarded benchmarks like BEND [r3], PEER [r4], and GUE [r5] have multiple datasets for only one or two tasks. Including multiple datasets for every task is often unnecessary and impractical, especially when carefully curated, representative datasets can already provide robust evaluations.
> >
> > - **Space for Future Improvements:** We wholeheartedly agree that benchmarking analyses can always be expanded and improved. While our current work lays a solid foundation by covering a wide range of tasks and datasets, we see this as an ongoing effort. Future iterations of our benchmark will incorporate more diverse datasets, refine task criteria, and address additional biological complexities to further strengthen its impact.
> >
> > We hope this clarifies our approach and demonstrates the balance we have struck between dataset diversity and the practical constraints of benchmark design. We would also like to invite you to consider evaluating our contribution in advancing the study of biological language models across different omics (DNA, RNA, and Protein), as reviewer 6UP5 has highlighted. Thank you again for your engagement and constructive suggestions, which we believe will help us improve our work and its impact.
> >
> > [r1] BEACON: Benchmark for Comprehensive RNA Tasks and Language Models. NeurIPS 2024.
> >
> > [r2] DART-Eval: A Comprehensive DNA Language Model Evaluation Benchmark on Regulatory DNA. NeurIPS 2024.
> >
> > [r3] BEND: Benchmarking DNA Language Models on biologically meaningful tasks. ICLR 2024.
> >
> > [r4] PEER: A Comprehensive and Multi-Task Benchmark for Protein Sequence Understanding. NeurIPS 2022.
> >
> > [r5] Dnabert-2: Efficient foundation model and benchmark for multi-species genome. ICLR 2024.

---

> ### Comment · Reviewer_8ScB · 2024-12-02
> **Thanks for your response, but I will keep my score**
>
> Thanks for your response, but they do not directly answer my questions or address my concerns. I believe that a benchmarking paper should use various datasets to increase the sample size and compare the model performance under different conditions to report an average effect. It has high risk to have a biased evaluation based on a single dataset. Since you mentioned that you performed a comprehensive benchmarking analysis, this point is even more important, which is the limit of a good benchmarking paper. I will raise the same questions for these benchmarking papers if they only include one dataset, and people will not trust it easily. Also, my question about information overlap is not well-addressed. I suggest the authors to investigate the biology background and discuss the shared gene elements in certain datasets, and explore if their effects are different or not.

---

> > ### Author Response · Authors · 2024-12-03
> > **Consider our benchmark scope**
> >
> > Thank you for your detailed response. We appreciate your concerns and would like to response more comprehensively.
> >
> > **Scope and Focus of Our Benchmark:** The primary goal of our work is to evaluate across different omics domains (DNA, RNA, and proteins). To mitigate bias, we have included multiple tasks for each omics domain. We kindly ask you to consider our contributions from this broader perspective of multi-omics benchmarking, rather than focusing solely on whether individual tasks have multiple datasets. This approach aligns with the design of other established biological benchmarks like BEACON (NeurIPS 2024), DART-Eval (NeurIPS 2024), BEND (ICLR 2024), PEER (NeurIPS), which also emphasize comprehensive scope rather than the inclusion of multiple datasets for every single task.
> >
> > **Tasks with Multiple Datasets:**  The sequences in the two types of enhancer activity prediction datasets (housekeeping and developmental promoters) overlap intentionally, as the focus is on evaluating the activity differences of the same sequences under these two conditions. Furthermore, even if the enhancer activity task is not strictly considered as having multiple datasets for a single task, we have also supplemented the Alternative Polyadenylation (APA) task with additional datasets to enhance the study of isoform diversity, including HSPE1, SNHG6, and WHAMMP2. Our current benchmark already encompasses 17 tasks, representing a substantial effort. In future iterations, we will take your suggestions into account and further expand the number of datasets for each task to ensure a more comprehensive evaluation from a biological perspective.

---

### Official Review · Reviewer_z8kJ · 2024-10-30

**Soundness:** 3
**Presentation:** 3
**Contribution:** 2
**Rating:** 5
**Confidence:** 4

**Summary:**

This paper introduces a comprehensive multi-omics benchmark encompassing a diverse collection of 17 cross-omics downstream tasks and datasets. It evaluates a set of state-of-the-art (SOTA) foundational language models, providing detailed descriptions of both implementation and outcomes. The project represents a significant amount of work and offers valuable information for the research community. The paper is well-written and easy to follow. While the paper excels as a resource, it lacks methodological novelty and the findings are largely intuitive. Addressing these areas and providing more in-depth analysis and discussion would significantly enhance the paper's contribution to the field.

**Strengths:**

Comprehensive Benchmark: The introduction of a multi-omics benchmark covering 17 diverse tasks and datasets is a significant contribution. It provides a standardized framework for evaluating the performance of various models across different omics data types.
Detailed Implementation: The paper includes thorough descriptions of the implementation process, which enhances reproducibility and allows other researchers to build upon this work.
Valuable Resource: The benchmark and the accompanying evaluations serve as a valuable resource for the community, facilitating future research and development in the field of multi-omics.

**Weaknesses:**

Methodological Novelty: While the paper is resource-rich, it lacks methodological innovation. The primary focus is on benchmarking existing models rather than introducing new techniques or approaches.
Insightfulness of Findings: Many of the findings presented are intuitive and do not offer deep insights into the underlying mechanisms or potential improvements. More in-depth analysis and discussion of the results would enhance the paper's impact.

**Questions:**

Benchmark Scope: While the benchmark is comprehensive, it would be beneficial to discuss any limitations or potential biases in the selection of tasks and datasets. This would provide a more balanced perspective and guide future expansions of the benchmark.
Comparison with Existing Benchmarks: A comparison with existing multi-omics benchmarks, if any, would help contextualize the contributions of this work and highlight its unique aspects.
Future Directions: The paper could benefit from a discussion on potential future directions, such as the integration of additional omics data types, the development of new evaluation metrics, or the exploration of novel model architectures.
Practical Applications: Including examples of practical applications or case studies where the benchmark has been used to derive meaningful biological insights would demonstrate the real-world utility of the resource.

---

> ### Author Response · Authors · 2024-11-24
> **Answer to Reviewer z8kJ**
>
> Thank you for the insightful feedback. We appreciate the opportunity to clarify and refine our work based on your review.
>
> ### **Weakness 1: Justifications for Methodological Novelty**
>
> We appreciate the reviewer’s observation regarding methodological novelty and would like to clarify the intent and scope of our work. We believe that a benchmark can make substantial contributions even without proposing a new method. Similar to benchmark studies such as PEER [r1] and ProteinGym [r2], COMET offers a novel analytical perspective to explore the capabilities of existing biological language models—particularly their ability to tackle tasks beyond the boundaries of their respective omics domains. Our standardized datasets and benchmarks ensure fair evaluation of architectures, algorithms, and training strategies and reveal model strengths and weaknesses that are overlooked in their original studies. As far as we know, our work is the first to address the gap in multi-omics studies by curating datasets and evaluating tasks, providing a strong foundation for further research.
>
> This paper is designed as an unbiased benchmark study to evaluate existing methods rather than using the benchmark to promote our own methods. Our goal is to establish a standardized benchmark that enables researchers to assess state-of-the-art methods objectively and as a foundation for future methodological development. Developing new methods will be the focus of future work and will be based on the insights gained from this study.
>
> [r1] PEER: A Comprehensive and Multi-Task Benchmark for Protein Sequence Understanding. NeurIPS 2022.
>
> [r2] ProteinGym: Large-Scale Benchmarks for Protein Design and Fitness Prediction. NeurIPS 2023.
> ### **Weakness 2: Explanation of Findings**
> We appreciate your valuable feedback. Our findings encompass a range of novel insights, which we categorize into two main types:
>
> 1. **Novel discoveries** (these insights can inspire researchers to develop new algorithms and conduct experimental analyses):
>
> - Randomly initialized vocabulary embeddings reveal cross-omics knowledge learned during pre-training.
> - Protein models enhance predictive performance in DNA and RNA regulatory tasks.
> - DNA models demonstrate potential in protein and RNA tasks, reflecting DNA's foundational role in the central dogma.
> - Single-omics models achieve competitive performance in their respective tasks, particularly structural tasks.
> - Multi-omics models outperform single-omics models on multi-molecular tasks.
> - Multi-molecular tasks remain significantly challenging and require further exploration.
> 2. **Findings aligned with existing work but expanded to broader contexts**:
>
> - CDS models demonstrate competitive performance on codon sequence data.
> - Nucleotide models have the potential to rival CDS models.
> (We extend findings similar to those in CaLM and "Are Genomic Language Models All You Need?" to more foundational models, including RNA models, making these discoveries more broadly applicable.)
>
>
> In response, we have reflected on your suggestions and revised the manuscript to enhance the depth and clarity of our findings. Key updates, highlighted in the manuscript `RESULTS` in `blue`, are as follows:
> - Enhanced Results Structure: We have reorganized and refined the results section to present experimental outcomes more logically and concisely. This includes adjusting the sequence of conclusions and improving the flow of their presentation.
> - Detailed Analysis: Additional explanations and context have been provided for the experimental results. For instance, we now discuss cross-omics adaptability, emphasizing how multi-omics models capture intricate molecular features by leveraging nucleotide, codon, and protein-specific representations. We also highlight the role of tailored pretraining on omics-specific data in achieving success.
> - Deeper Insights: To make conclusions more insightful and actionable, we analyzed how nucleotide models implicitly learn codon patterns and adapt to cross-molecular tasks, demonstrating the potential of unified multi-omics representations. This insight underlines the need for architectural innovations and task-specific adaptations, particularly for tasks requiring highly specialized knowledge.
> - Constructive Summaries: We added summary sections after the experimental analyses, offering clear and constructive takeaways. For example, the potential of multi-omics models to outperform task-specific models in capturing cross-omics dependencies is discussed, alongside the challenges faced by models like LucaOne in highly specialized tasks.
>
> These revisions aim to enhance the manuscript's impact by providing deeper insights, actionable conclusions, and a clearer understanding of the results. Thank you for encouraging us to improve the quality of our work.

---

> > ### Author Response · Authors · 2024-11-24
> > **Answer to Reviewer z8kJ**
> >
> > ### **Question 1: Discussion on Benchmark Scope**
> > Our principles for selecting tasks and datasets prioritize comprehensiveness and impactfulness. Comprehensiveness is ensured by including tasks and datasets across each omic type, their pairwise interactions, and biologically critical aspects in structure, function, and engineering. Impactfulness is achieved by selecting tasks and datasets from sources that are widely recognized as authoritative, peer-reviewed, and published in high-impact conferences or journals, as well as public datasets released in competitions.
> >
> > Based on these criteria, the limitations of our selection are as follows:
> > - We mainly focus on multi-omics sequence understanding tasks, there may be other types of data that can be expanded.
> > - Tasks involving interactions among more than two omics are not included.
> > - Biologically critical tasks and datasets that are not yet widely acknowledged or published in high-impact venues may be excluded.
> > - Informative but private datasets, which we do not have access to, are not considered.
> >
> > We believe these limitations reflect the trade-offs made to ensure the benchmark remains comprehensive, impactful, and grounded in publicly available, well-regarded resources.
> >
> >
> >
> > ### **Question 2: Comparision with Existing Benchmarks**
> > There is limited work exploring multi-omics benchmarks. To the best of our knowledge, the closest related work is Boshar et al.’s [r1] analysis of gLMs applied to protein-related tasks. They established a benchmark for four state-of-the-art (SOTA) models over five common protein tasks, curating CDS sequences to enable fair comparisons between pLMs and gLMs. However, their focus remains largely on single-omics tasks, emphasizing codon-to-protein mappings without addressing the broader scope of multi-omics interactions.
> >
> > Our benchmark (COMET) includes 11 models (8 foundation models and 3 naive models) on a total of 17 tasks across multiple omics. COMET distinguishes itself through its comprehensive multi-omics focus, extending beyond codon-to-protein tasks to incorporate genomics, transcriptomics, and proteomics. This integration allows COMET to evaluate cross-omics interactions, such as antibody-antigen neutralization and RNA-protein interaction, addressing complex biological challenges that single-omics benchmarks cannot capture. COMET’s unique strength lies in its ability to assess cross-omics tasks and capture biological interdependencies across interconnected omics layers. By filling the gap in multi-omics benchmarking, COMET supports the development of integrative models capable of addressing the inherent complexity of biological systems. This positions COMET as a critical resource for advancing research in systems biology and multi-omics integration.
> >
> >
> > [r1] Are genomic language models all you need? exploring genomic language models on protein downstream tasks. Bioinformatics 2024.
> > ### **Question 3: Future Directions**
> > We appreciate the reviewer’s suggestion and agree that future directions are crucial for expanding the impact of this work. We propose three potential areas for further exploration, highlighted in the manuscript `APPENDIX A.8` in `orange`.
> > - Incorporating Additional Omics Data and Biological Priors:
> > Expanding the benchmark to include additional omics data types and features that indirectly influence biological processes in downstream tasks will enhance its comprehensiveness. Integrating biological priors, such as pathway-level annotations, protein-protein interactions, or chromatin accessibility maps, can provide a more holistic view of molecular interactions and improve the relevance of downstream predictions.
> > - Refining the Evaluation Pipeline for Multimodal Data:
> > Developing a more sophisticated evaluation pipeline that better infuses multimodal data will be essential. For instance, metrics that capture cross-modality consistency and assess how well models leverage complementary information from multiple omics types can provide deeper insights into model performance. Additionally, incorporating metrics that account for the stochasticity and uncertainty inherent in biological systems can improve evaluation robustness.
> > - Developing Novel Multi-Omics Foundation Models:
> > Leveraging the insights gained from this benchmark, we aim to explore novel model architectures tailored for multi-omics data. These models could employ advanced techniques like attention-based integration and hierarchical representations of omics modalities. The benchmarking insights will guide the design of these models, ensuring they address the specific challenges and opportunities identified in multi-omics tasks.
> >
> > These future directions aim to expand the scope and utility of the benchmark, driving the development of innovative methods and fostering deeper biological insights through more accurate and comprehensive modeling of multi-omics data.

---

> ### Author Response · Authors · 2024-11-24
> **Answer to Reviewer z8kJ**
>
> ### **Question 4: Case Studies of Biology Benchmarks for Practical Applications**
> Benchmarks in biology have demonstrated their utility in uncovering meaningful biological insights by systematically evaluating different deep learning models.
>
> - For example, in gene expression prediction, studies by Sasse et al. [r1] and Huang et al. [r2] exposed generalizability issues in models like Enformer, Basenji2, ExPecto, and Xpresso, which underperformed in vivo. Their analysis revealed that Enformer overly relied on specific SNVs for predicting expression levels. Similarly, Khan et al.’s [r3] independent evaluation of scBERT demonstrated that while it generalizes well to new datasets, its performance is highly sensitive to class imbalance.
> - For multi-omics tasks, our benchmark can facilitate research in codon optimisation to boost protein yields [r4] and vaccine design [r5] in the real world. And it can also facilitate drug discovery, such as siRNA drugs [r6] for gene silencing and the design of more affinitive antibodies [r7].
>
> These benchmarks have been instrumental in identifying model strengths and limitations, providing actionable insights that drive improvements in architecture and training strategies. However, the absence of standardized benchmarks in multi-omics research continues to hinder the advancement in this field. Aiming to address this gap, our multi-omics benchmark provides a crucial foundation that systematically assesses existing models, uncovering their potential and biological insights across DNA, RNA and proteins.
>
>
> [r1] Benchmarking of deep neural networks for predicting personal gene expression from DNA sequence highlights shortcomings. Nature Genetics 2023.
>
> [r2] Personal transcriptome variation is poorly explained by current genomic deep learning models. Nature Genetics 2023.
>
> [r3] Reusability report: Learning the transcriptional grammar in single-cell RNA-sequencing data using transformers. Nature Machine Intelligence 2023.
>
> [r4] High-throughput 5′ UTR engineering for enhanced protein production in non-viral gene therapies. Nature Communications 2021.
>
> [r5] Algorithm for optimized mRNA design improves stability and immunogenicity. Nature 2023.
>
> [r6] On the art of identifying effective and specific siRNAs. Nature Methods 2006.
>
> [r7] AIntibody: an experimentally validated in silico antibody discovery design challenge. Nature Biotechnology 2024.

---

> ### Comment · Reviewer_z8kJ · 2024-11-26
>
> Thanks for your response which clarifies some of my concerns. I will keep my score.

---

> > ### Author Response · Authors · 2024-11-27
> > **Thank you for your response.  We want to address your remaining concerns**
> >
> > Thank you for your response. We deeply appreciate your feedback, and your insights will be valuable in helping us improve our work.  We are genuinely committed to addressing your concerns and engaging in meaningful discussions to resolve them.

---

> > ### Author Response · Authors · 2024-12-01
> > **Could you kindly share what your remaining concerns are?**
> >
> > Could you kindly share what your remaining concerns are? We would be happy to address them further to ensure our work meets your expectations.

---

### Official Review · Reviewer_TFBg · 2024-11-02

**Soundness:** 2
**Presentation:** 2
**Contribution:** 2
**Rating:** 5
**Confidence:** 4

**Summary:**

This paper presents the comprehensive multi-omics benchmark COMET (Benchmark for Biological Comprehensive Multi-omics Evaluation Tasks and Language Models), created to assess models across single-omics, cross-omics, and multi-omics tasks. The goal of this benchmark is to identify key challenges in multi-omics research and to guide future efforts, ultimately fostering advancements in understanding biological processes through the integrated analysis of diverse omics data.

**Strengths:**

This paper curated a collection of datasets and tasks covering structural and functional aspects in DNA, RNA, and proteins, including tasks that span multiple omics levels. This paper further evaluated a variety of FMs for respective Bio-modalities, offering insights into their performance, especially with respect to cross-modality applications.

**Weaknesses:**

The introduction to the various benchmark tasks is thorough. However, it would be advantageous to provide more detailed information about the AIML models being evaluated, particularly regarding their potential strengths and weaknesses for specific tasks. Additionally, it is recommended to explain other aspects of model training and evaluation, such as the criteria for choosing between LoRA fine-tuning and full fine-tuning for each model, and the rationale behind selecting specific metrics for evaluating each model.

Results interpretation is rather brief and sometimes confusing.

•	What is the key takeaway from all the experiments when comparing Literature SOTA to Pre-trained FMs? Specifically, Literature SOTA outperformed in all protein-related tasks listed in Table 3—what insights can be drawn from this? Additionally, no RNA-based FMs achieved top performance in any RNA tasks—what insights can be provided here? Could it be related to how much data was used in pre-training the RNA-based FMs? The summary text in section 5.2 does not accurately reflect the data presented in Table 3, which is causing confusion.

•	For 5.3, CROSS-MOLECULAR BENCHMARK RESULTS, why the performance on EC is so much worse for CaLM after refinement, which is typically not the case if refinement is properly done? Also for EC task, using condon sequence gets noticeably worse results compared to its protein sequence counterpart. What is special about EC task compared to other tasks like Beta-Lac and Flu, which might contribute to this difference?

•	For 5.4 MULTI-MOLECULAR BENCHMARK RESULTS, In contrast to single-molecular tasks, for multi-molecular tasks, Literature SOTA still dominantly work better compared to multi-omics models or combination of two single-omics models. It would be important to elaborate the possible limitations in the current implementation of the multi-omics models that leads to this contrast and inferior performance.

The related work of paper can be strengthened and some of the claims can be formulated in the appropriate context of existing works. For example, the paper claims to be the first to establish a benchmark for "compiling tasks and data involving cross-molecules and multi-molecules" and "evaluate existing foundation models for DNA, RNA, and proteins, as well as multiomics approaches. We conduct experiments with fully-finetuned or frozen models." Recent works such as Prakash, Moskalev et al., 2024 (Bridging biomolecular modalities for knowledge transfer in bio-language models) and Boshar et al., 2024 (Are Genomic Language Models All You Need?) should be cited as laying the foundations for these ideas and uncovering the link to central dogma which the paper claims as their contribution.

Some of the conclusions have already been derived in prior works such as "CDS model demonstrates competitive performance on codon sequence data" has been reported already in CaLM (Outerial & Deane, 2024)

**Questions:**

Provided in Weakness section

---

> ### Author Response · Authors · 2024-11-24
> **Answer to Reviewer TFBg**
>
> We sincerely appreciate Reviewer TFBg's thoughtful feedback, and here we provide corresponding responses to address these concerns.
>
> ### **Weakness1: Details About Foundation Models**
>
> Thank you for your suggestions. To better facilitate reader understanding, we have added more detailed information about the foundational biological models we used in the manuscript `APPENDIX A.6` in `blue`. The revisions primarily involve two aspects.
> - We have provided more detailed information on the data and tasks involved in the pre-training process of different foundational models.
> - We have conducted an analysis of the potential strengths and weaknesses of different models.
>
> All pre-trained biology foundation models utilize a Masked Language Modeling (MLM) objective.
>
> - DNABERT2 utilized datasets from the human genome and multiple species genomes, totaling 35.24 billion nucleotide bases. The human genome dataset comprised 2.75 billion nucleotide bases, while the multi-species genome dataset included 32.49 billion nucleotide bases from the genomes of 135 different species. During the data processing, all sequences containing 'N' were removed, leaving only those composed of ATCG nucleotides.
>
> - NTv2 leverages three datasets: the Human reference genome dataset, which contains 3.2 billion nucleotides; the 1000 Genomes Project (1000G) dataset, featuring over 20.5 trillion nucleotides; and the Multispecies dataset, comprising 174 billion nucleotides from 850 species. During the preprocessing phase, all nucleotides outside of ATCG are replaced with 'N'. For both the multispecies and human reference datasets, the genomes are segmented into overlapping chunks of 6,100 nucleotides. Each chunk overlaps with the previous one by sharing the first 50 nucleotides and with the next one by sharing the last 50 nucleotides.
>
> - The RNA-FM model is pre-training utilizing data sourced from RNACentral. To ensure the non-redundancy of the dataset, RNA-FM employs CD-HIT (specifically, CD-HIT-EST) with a threshold set at 100\% sequence identity. This process led to a final dataset comprising 23.7 million distinct RNA sequences.
>
> - The BEACON-B model uses 523,934 human ncRNA sequences filtered from the total ncRNA in the RNACentral database as pre-training data.
>
>
> - The ESM-1b model was pre-trained on Uniref50, which comprises approximately 30 million protein sequences. During the training process, sequences exceeding 1023 tokens (excluding the CLS token) are randomly truncated to a length of 1023 tokens.
>
> - The ESM-2 model is trained using the UniRef50 dataset. To enhance the data volume and diversity, during each training update, a mini-batch of sequences from UniRef50 is sampled and replaced with sequences uniformly sampled from the corresponding UniRef90 clusters. This approach allows the ESM-2 model to be trained on over 60 million protein sequences.
>
> - The CaLM model is pre-trained using cDNA data collected from the European Nucleotide Archive database. During preprocessing, sequences containing unknown nucleotides, start codons that are not ATG, internal stop codons, or a nucleotide count not divisible by three are removed. The final dataset consists of about 9 million cDNA sequences.
>
> We can see that all models are trained using MLM, among which the NTv2 model uses much more pre-training data than other models, which may make it more generalizable on more tasks. The RNA-based models are all pre-trained only on ncRNA, and BEACON-B only uses a small part of the pre-training data, which may affect their potential performance. Models like the protein-based model and the Calm codon model use amino acid (non-overlapping 3mer) encoding, which may have an advantage in tasks that focus on amino acid expression. For the RNA-FM and ESM-1b models, the use of absolute position encoding limits the length of their input sequences, which may affect their performance on long sequence tasks.
>
> ### **Weakness2: Fine-tuning Strategy Selection Criteria**
> - We only utilize LoRA fine-tuning for the large model（LucaOne, which has about 1B parameters）to obtain a fair comparison with full fine-tuning as possible. It not only conserves computational resources but also ensures good performance of the model on downstream tasks.
> - We employ full fine-tuning for all other models to fully leverage the potential of these models and ensure optimal results across various tasks.

---

> ### Author Response · Authors · 2024-11-24
> **Answer to Reviewer TFBg**
>
> ### **Weakness3: Rationale Behind Selecting Metrics**
>
> To ensure fairness and consistency, we maintain the evaluation metrics that are consistent with the original task, facilitating alignment with the performance of the original task [r1,r2,r3]. For the evaluation metrics selection, this paper encompasses three categories of 17 tasks: Single-omics, Cross-molecule, and Multi-molecules, with task objectives involving structure, function, and engineering design. The task types include single-label regression, multi-label regression, and multi-label classification, with test set sample sizes ranging from 40 to 49,755.  And like other bio-benchmark work, such as the PEER benchamrk (NeurIPS 2022) [r2] also has different metrics for regression tasks such as SCC and RMSE, the BEACON benchmark (NeurIPS 2024) [r4] uses different metrics for regression tasks such as SCC, R^2, and RMSD, and similarly, the GUE benchamrk (ICLR 2024) [r5] uses different metrics for classification tasks such as MCC and F1.
>
>
> [r1] SAPROT: PROTEIN LANGUAGE MODELING WITH STRUCTURE-AWARE VOCABULARY. ICLR 2024.
>
> [r2] PEER: A Comprehensive and Multi-Task Benchmark for Protein Sequence Understanding. NeurIPS 2022.
>
> [r3] Are genomic language models all you need? exploring genomic language models on protein downstream tasks. Bioinformatics 2024.
>
> [r4] BEACON: Benchmark for Comprehensive RNA Tasks and Language Models. NeurIPS2024.
>
> [r5] Dnabert-2: Efficient foundation model and benchmark for multi-species genome. ICLR 2024.
>
> ### **Weakness4:  Results and Findings Interpretation**
>
>
> We appreciate your valuable feedback. Our findings encompass a range of novel insights, which we categorize into two main types:
>
> 1. **Novel discoveries** (these insights can inspire researchers to develop new algorithms and conduct experimental analyses):
>
> - Randomly initialized vocabulary embeddings reveal cross-omics knowledge learned during pre-training.
> - Protein models enhance predictive performance in DNA and RNA regulatory tasks.
> - DNA models demonstrate potential in protein and RNA tasks, reflecting DNA's foundational role in the central dogma.
> - Single-omics models achieve competitive performance in their respective tasks, particularly structural tasks.
> - Multi-omics models outperform single-omics models on multi-molecular tasks.
> - Multi-molecular tasks remain significantly challenging and require further exploration.
> 2. **Findings aligned with existing work but expanded to broader contexts**:
>
> - CDS models demonstrate competitive performance on codon sequence data.
> - Nucleotide models have the potential to rival CDS models.
> (We extend findings similar to those in CaLM and "Are Genomic Language Models All You Need?" to more foundational models, including RNA models, making these discoveries more broadly applicable.)
>
>
> In response, we have reflected on your suggestions and revised the manuscript to enhance the depth and clarity of our findings. Key updates, highlighted in the manuscript `RESULTS` in `blue`, are as follows:
> - Enhanced Results Structure: We have reorganized and refined the results section to present experimental outcomes more logically and concisely. This includes adjusting the sequence of conclusions and improving the flow of their presentation.
> - Detailed Analysis: Additional explanations and context have been provided for the experimental results. For instance, we now discuss cross-omics adaptability, emphasizing how multi-omics models capture intricate molecular features by leveraging nucleotide, codon, and protein-specific representations. We also highlight the role of tailored pretraining on omics-specific data in achieving success.
> - Deeper Insights: To make conclusions more insightful and actionable, we analyzed how nucleotide models implicitly learn codon patterns and adapt to cross-molecular tasks, demonstrating the potential of unified multi-omics representations. This insight underlines the need for architectural innovations and task-specific adaptations, particularly for tasks requiring highly specialized knowledge.
> - Constructive Summaries: We added summary sections after the experimental analyses, offering clear and constructive takeaways. For example, the potential of multi-omics models to outperform task-specific models in capturing cross-omics dependencies is discussed, alongside the challenges faced by models like LucaOne in highly specialized tasks.
>
> These revisions aim to enhance the manuscript's impact by providing deeper insights, actionable conclusions, and a clearer understanding of the results. Thank you for encouraging us to improve the quality of our work.

---

> ### Author Response · Authors · 2024-11-24
> **Answer to Reviewer TFBg**
>
> ### **Weakness5: Comparing Literature SOTA to Pretrained FMs**
>
> - Similar results that the pre-trained model is lower than the literature SOTA can also be observed in PEER benchmark [r4]. Literature SOTA models for specific downstream tasks can incorporate task-specific priors and post-processing. For example, in protein-related tasks SOTA, MSATrans introduces multiple sequence alignment (MSA), integrating evolutionary information related to proteins [r1]. SaProt incorporates a spatial structure vocabulary, combining tertiary structure information to aid functional prediction [r2]. In contrast, language models focus on the generalizability of sequence representations and do not include such task-specific information, which are orthogonal. For example, incorporating priors or additional features into language models can enhance task performance [r5]. However, this study primarily focuses on evaluating the representational capabilities of biological foundation models.
>
> - Furthermore, Literature SOTA model ESM-1v [r3], as a general protein language model, uses 98 million diverse protein sequences, significantly exceeding the data volume and diversity of ESM-1b. It achieves SOTA performance in protein Thermostability downstream task, demonstrating that increasing training data enhances the overall performance of language models and highlighting the significant potential of general biological language models.
>
> [r1] MSA Transformer. ICML 2021.
>
> [r2] SAPROT: PROTEIN LANGUAGE MODELING WITH STRUCTURE-AWARE VOCABULARY. ICLR 2024.
>
> [r3] Language models enable zero-shot prediction of the effects of mutations on protein function. NeurIPS 2021.
>
> [r4] Peer: a comprehensive and multi-task benchmark for protein sequence understanding. NeurIPS 2022.
>
> [r5] Predicting Antimicrobial Peptides Using ESMFold-Predicted Structures and ESM-2-Based Amino Acid Features with Graph Deep Learning. JCIM 2024.
>
> ### **Weakness6: Performance on RNA Tasks**
>
> Regarding the issue that RNA foundation models have not achieved the top performance in any RNA tasks, we think the factors may originate from three aspects:
>
> - **Pre-training Data Scale:** RNA-FM is pre-trained on approximately 27 million sequences, while BEACON uses a much smaller subset of this data (only about 0.5 million sequences) to achieve efficient and cost-effective RNA models. In comparison, ESM-1b, which has a similar training scale, uses around 30 million protein sequences and performs worse than RNA foundation models on RNA downstream tasks. Models with better performance, such as ESM-2 and DNABERT2, are trained on nearly 60 million sequences. NTv2, which has superior comprehensive performance, is trained on an even larger dataset. This suggests that the scale of pre-training data can significantly impact performance on downstream tasks.
>
> - **Omics Type of Pre-training Data Type:** It is important to note that both RNA-FM and BEACON are trained using data from the RNACentral database, which primarily includes non-coding RNA. The exclusion of coding RNA might result in incomplete information learned by the models, potentially affecting their performance on downstream tasks. We observe that DNA models, while performing well on RNA property tasks, underperformed on RNA structure tasks. In contrast, protein models showed good generalization across various RNA tasks. We hypothesize that, based on the central dogma, DNA, RNA, and proteins share many similarities in their properties and functions, making knowledge transfer relatively easy. However, DNA sequences are less expressive of spatial structure information compared to proteins. Therefore, protein pre-trained models, which learn more structural information, generalize better to spatial structure tasks compared to DNA models.

---

> ### Author Response · Authors · 2024-11-24
> **Answer to Reviewer TFBg**
>
> ### **Weakness7: Explanation of EC Task**
>
> Regarding the performance changes of CaLM on the EC task, we first check whether the fine-tuning was correctly completed. When using the CaLM model for downstream task predictions, we directly utilize the official code and weights(from  https://github.com/oxpig/CaLM) and add only a linear layer for adaptation to the downstream task. The code is reviewed by two individuals to ensure its correctness. In order to ensure the reliability of the experimental results, we tried different hyperparameters，and the results are shown in the table.
>
>
> | learning_rate  |  0.001 | 0.0001 | 0.0005 | 0.00001 | 0.00005 | 0.000001 | 0.000002 | 0.000005 |
> |:--------------:|:------:|:------:|:------:|:-------:|:-------:|:--------:|:--------:|:--------:|
> |  Frozen Finetuning  | 0.7494 | 0.6752 | 0.7337 |  0.4434 |  0.6203 |  0.1431  |  0.1427  |  0.3401  |
> |       Full Parameter Finetuning      | 0.1777 | 0.5006 |  0.445 |  0.4423 |  0.489  |  0.3532  |  0.4097  |  0.4335  |
>
>
> During the process of adjusting different learning rates, we observed that fine-tuning indeed led to a decline in CaLM's performance on the EC task. We speculate that the possible reason for the performance degradation of CaLM on the EC task could be:
>
> - In the other two downstream tasks of proteins, the sequences will involve a large number of variant non-natural sequences, while the sequences of the EC task are mainly natural protein sequences, which are closer to the data distribution learned by pre-training. Fine-tuning all parameters may make it easier for Calm to forget the knowledge learned in pre-training, while the frozen method can slow down the forgetting of Calm pre-training knowledge to a certain extent.
>
>
> ### **Weakness8: Using Protein Sequence Better Than Codon Sequence**
>
> Regarding the issue of using amino acid sequences outperforming using nucleotide sequences, we find that the Flu task shows little difference when using amino acid sequences versus nucleotide sequences, whereas the Beta-Lac and EC tasks perform significantly better with amino acid sequences compared to nucleotide sequences. A possible explanation is that the Flu task involves predicting sequence properties, where single-nucleotide mutations leading to amino acid variations (even synonymous codons resulting in the same amino acid) can alter the sequence properties. This aligns with the granularity of single nucleotides. In contrast, the Beta-Lac and EC tasks involve changes at the codon level. Beta-Lac focuses on property changes due to codon mutations, while EC examines how amino acid changes affect protein function. Therefore, the tokenization methods used by DNA and RNA models (such as BPE, non-overlapping 6mer, and single) provide good generalization at the single-nucleotide level but perform relatively poorly on tasks that emphasize the importance of non-overlapping 3mers, compared to directly modeling amino acid sequences.
> ### **Weakness9: Limitations of Multi-Omics Foundation Models**
>
> Current multi-omics models still have room for improvement, with the main limitations being the following:
>
> - Simple Concatenation for Single-Omics Model Integration: Current representation fusion methods rely on very simple concatenation techniques, which limit the exploration of information associations between different omics.
> - Lack of Biological Prior Knowledge and Task-Specific Optimization: Current multi-omics methods often directly adopt the transfer learning approach from language models, focusing more on the generalizability of sequence representations. They do not incorporate prior biological knowledge and lack specific optimizations or post-processing tailored to downstream tasks.
> - Primitive Pre-training Methods for Multi-Omics Data: The pre-training methods for different omics data in current multi-omics models are still quite basic. We observe consistently lower performance in experiments involving proteins, which may indicate insufficient learning of protein representation knowledge.

---

> ### Author Response · Authors · 2024-11-24
> **Answer to Reviewer TFBg**
>
> ### **Weakness10: More Appropriate Context of Existing Works**
>
> Thank you for your feedback and suggestions. We have improved the context and introduction of the relevant work in the manuscript `RELATED WORK` in `blue`.
>
> Recent studies, such as "Are Genomic Language Models All You Need?" [r1] and "Bridging Biomolecular Modalities for Knowledge Transfer in Bio-Language Models" [r2], have explored the idea of transferring pre-trained biological models to other omics domains. Researching multi-omics is a cutting-edge and popular topic in this field, aiming to integrate and utilize information from various types of biological data to gain deeper insights into complex biological systems.
>
> - We have already included "Are Genomic Language Models All You Need?" in the related works section of our initial manuscript and greatly value its contributions.
> - Although our work (submission for ICLR deadline on October 1, 2024) predates the work "Bridging Biomolecular Modalities for Knowledge Transfer in Bio-Language Models" (bioRxiv submission on October 17, 2024), we will follow your kind suggestion and incorporate this work into the revised version of our manuscript.
>
> To make our claim more clear, we have reformulated our contribution as: "We present the first benchmark that encompasses single-omics tasks, cross-omics tasks, and multi-omics tasks spanning DNA, RNA, and protein sequences." By establishing this benchmark, we hope to provide a standardized platform for comparing and improving the capabilities of multi-omics models, thereby promoting more comprehensive and accurate biological research.
>
> [r1] Are genomic language models all you need? exploring genomic language models on protein downstream tasks. Bioinformatics 2024.
>
> [r2] Bridging biomolecular modalities for knowledge transfer in bio-language models. bioRxiv 2024.
>
> ### **Weakness11: Similar Findings with CaLM**
>
> - CaLM is an outstanding contribution to the field, and we recognize and highly commend its work. In the process of establishing a benchmark for multi-omics frameworks, we arrived at the same conclusion as CaLM: "CDS model demonstrates competitive performance on codon sequence data."
> - The study "Are genomic language models all you need? Exploring genomic language models on protein downstream tasks" [r1] reached similar conclusions after CaLM and was published in Bioinformatics. Moreover, we further explored the potential of nucleotide models, including RNA models, to excel in protein-related tasks, making this finding more universally applicable. This highlights how our benchmark not only reinforces such discoveries but also drives impactful advancements within the community—just one of the many contributions our benchmark offers.
>
> [r1] Are genomic language models all you need? exploring genomic language models on protein downstream tasks. Bioinformatics 2024.

---

> ### Comment · Reviewer_TFBg · 2024-11-27
>
> Author's rebuttal and editing of the manuscript partially addressed my questions, in experiment setup, validation, and interpretation. To draw more insightful conclusion, a more rigorous experiments (e.g. FM models with different/comparable sizes, consistent finetuning schemes, data nature/quality of evaluated tasks and their influence etc) are desired, but may not be feasible during the limited rebuttal timeframe. I have increased my score to 5.

---

> ### Author Response · Authors · 2024-12-01
> **Thanks for your feedback**
>
> Thank you for your thoughtful feedback and for increasing your score. We appreciate your acknowledgment of the progress made in addressing your concerns.
>
> To further address your suggestion regarding model sizes, we conducted additional experiments exploring the impact of model scale. Specifically, we evaluated NTv2 and ESM-2 models of varying sizes across DNA, RNA, and protein tasks.
>
> - **For tasks aligned with the model's corresponding omics domain,** such as NTv2 on DNA and RNA tasks and ESM-2 on protein tasks, the results indicate that larger model sizes generally lead to improved performance, indicating task-specific scaling benefits.
>
> - **For other omics tasks,** we observed that even a smaller model like ESM-2_8m can achieve performance comparable to the larger NTv2 model. We hypothesize that this may be attributed to the significantly larger volume of pretraining sequences available for ESM-2 on protein data compared to NTv2 on DNA data. This insight could guide future research on pretraining nucleotide models with larger datasets for enhanced cross-omics capabilities.
>
> We hope these findings add further clarity and demonstrate our commitment to enhancing the rigor and depth of our benchmarking analysis. Please let us know if there are any additional aspects you would like us to address.
> |                                         | DNA      |         | RNA   | Protein      |
> |-----------------------------------------|----------|---------|-------|--------------|
> |                Model/Task               | EA (Dev) | EA (Hk) |  APA  |     Ther     |
> |                  Metric                 |    PCC   |   PCC   |  R^2  | Speraman's p |
> | Pretrained Omics Language Model(Frozen) |          |         |       |              |
> |                 NTv2_50m                |   30.24  |  30.64  | 26.47 |     61.09    |
> |                NTv2_100m                |   29.36  |  28.89  | 23.09 |     60.95    |
> |                NTv2_250m                |   36.44  |  37.99  | 31.92 |     61.45    |
> |                NTv2_500m                |   32.80  |  35.22  | 28.86 |     58.90    |
> |                                         |          |         |       |              |
> |                 ESM-2_8m                 |   48.33  |  65.13  | 38.16 |     61.15    |
> |                 ESM-2_35m                |   47.84  |  61.57  | 45.79 |     63.07    |
> |                ESM-2_150m                |   43.27  |  54.73  | 35.51 |     63.02    |
> |                ESM-2_650m                |     -    |    -    | 43.57 |     64.76    |

---

> > ### Comment · Reviewer_TFBg · 2024-12-02
> >
> > Thanks for one additional experiment related to model size. I will keep my (already increased) score, considering for a benchmark paper on different models, more rigorous and comprehensive experiment with in-depth understanding / insights / variation with respect to the underlying model being evaluated is important.

---

> > > ### Author Response · Authors · 2024-12-03
> > > **Finetuning schemes and data nature/quality**
> > >
> > > **1. Finetuning Schemes**
> > > To further address your suggestions on fine-tuning strategies, we add additional experiments to explore the impact of lora and frozen. For comprehensive evaluation, we select one task from DNA, RNA, and protein for each experiment.
> > >
> > > - We observed that in most cases, the performance of the LoRA fine-tuning strategy is better than frozen and close to all. This shows that LoRa as a PEFT method can achieve relatively good results at a low cost, and further proves the rationality of our experimental setting.
> > > - On the other hand, we found that frozen had less loss in protein tasks than lora. We speculate that this may be because people have studied protein pre-training models more thoroughly, and the pre-training weights contain more possibly correct information. It also shows that there is still great potential for the development of DNA and RNA pre-training models.
> > >
> > >
> > > We hope that additional experiments will make the entire article more rigorous.
> > >
> > > |   Model  | Enhancer Activity(Dev) |        |       | Enhancer Activity(Hk) |        |       | Programable RNA |        |       | Thermostability |        |       |
> > > |:--------:|:----------------------:|:------:|:-----:|:---------------------:|:------:|:-----:|:-------------:|:------:|:-----:|:---------------:|:------:|:-----:|
> > > |          |           all          | frozen |  lora |          all          | frozen |  lora |      all      | frozen |  lora |       all       | frozen |  lora |
> > > | DNABERT2 |          68.22         |  39.44 | 66.32 |         77.43         |  41.75 | 75.93 |     54.79     |  19.99 | 54.24 |      16.54      |  60.94 | 53.27 |
> > > |   NTv2   |          66.2          |  29.36 | 65.98 |         76.51         |  28.89 | 75.85 |     55.27     |  23.09 | 54.86 |      60.19      |  60.95 | 57.42 |
> > > |   RNA-FM  |          68.87         |  36.31 | 69.03 |         77.76         |  37.86 | 77.38 |     55.98     |  20.05 | 57.36 |      55.31      |  56.8  | 40.32 |
> > > |  BEACON-B  |          66.05         |  34.57 | 66.86 |         76.31         |  38.63 | 76.39 |     54.67     |  25.73 | 52.21 |      60.76      |  57.18 | 50.98 |
> > > |   ESM-1b  |          62.21         |  51.71 | 67.06 |         73.81         |  63.31 | 76.54 |     54.42     |  52.29 | 56.32 |      70.94      |  69.83 | 70.58 |
> > > |   ESM-2   |          68.04         |  43.27 | 68.82 |         77.03         |  54.73 |  77.5 |     56.27     |  35.51 | 56.94 |      69.36      |  63.02 | 65.87 |
> > >
> > > **2 Data Nature of Tasks**
> > > * Gene Expression
> > >     We adopt the data processing methodology from Xpresso[r1]. Human gene expression data comes from the Epigenomics Roadmap Consortium, which provides normalized RNA-seq values for protein-coding mRNAs across 56 tissues and cell lines.
> > >
> > >     Due to the large number of parameters in biological language models and the memory limitations of A100 GPUs, our experiments show that trimming sequence lengths to 6000 bp ensures compatibility with all models for processing input sequences. By inputting consecutive 6000 bp nucleotide fragments from different positions in the processed sequences into the Xpresso model, we identify that the sequence indexed from position 7000 to 12999 (length 6000 bp) achieves optimal test performance. This segment contains the most information related to gene expression levels.
> > >
> > >     For training, we use the 6000 bp nucleotide sequence indexed from position 7000 to 12999 as input and the expression data for 56 tissues as labels. The train, validation, and test dataset splits follow the methodology used in Xpresso.
> > >
> > > * Enhancer Activity Prediction
> > >     We follow the processing procedure described in [r2]. The data includes sequence information and transcriptional activity metrics for both Drosophila and humans, encompassing developmental and housekeeping transcriptional activity levels.
> > >
> > >     We use downloaded sequences of 249 bp in length, along with `Dev_log2_enrichment_scaled` and `Hk_log2_enrichment_scaled`, which respectively represent developmental and housekeeping transcriptional activity information. The dataset is divided into training, validation, and test sets according to the method outlined in [r2].
> > >
> > >
> > > [r1] Predicting mrna abundance directly from genomic sequence using deep convolutional neural networks. Cell Reports 2020.
> > >
> > > [r2] DeepSTARR predicts enhancer activity from DNA sequence and enables the de novo design of synthetic enhancers. Nature Genetics 2022.

---

> > > > ### Author Response · Authors · 2024-12-03
> > > > **Finetuning schemes and data nature/quality**
> > > >
> > > > * APA Isoform Prediction
> > > >     The preparation for IPA isoform analysis begins by filtering raw sequencing reads from all MPRAs[r3] to retain only high-quality, full-length RNA sequences. These reads are grouped based on the randomized regions located upstream of the proximal polyadenylation site (pPAS), forming a dictionary of sequence variants for each library. To expand this dictionary, sequencing is also performed on the plasmid library, capturing members that lack expression of a distal isoform. RNA reads are then matched to dictionary entries by identifying the upstream region with the shortest Hamming distance.
> > > >
> > > >     Polyadenylation cleavage sites are determined for each mapped read by detecting the presence of a Poly-A tail. The cleavage positions are recorded as vectors associated with individual sequence variants, including a specific position for reads mapping to non-random distal sites. The dataset generated from this process consists of a dictionary of distinct sequence variants paired with vectors of cleavage position counts. A final filtering step ensures data quality by discarding sequences supported by fewer than 10–20 unique UMI RNA reads or those containing over 75\% A-nucleotides within a 12–20 bp region, which could indicate internal priming artifacts.
> > > >
> > > > We process data from 12 random 3' UTR libraries. 9 among the 12 libraries are used for training and 3 held out  (the 3 held-out libraries were excluded from the current analysis). To construct a balanced test set, sequences from each library are first shuffled independently according to their read counts. These shuffled sequences are then merged using a round-robin approach, selecting one sequence from each library at a time in descending order of read count. This strategy ensures that the test set contains an even representation of high-read count sequences across all libraries. The remaining sequences are appended to the beginning of the combined library, and the training set is further shuffled to enhance randomness. For benchmarking purposes, the top 10\% of high-read count sequences are prioritized. Among these, the most abundantly expressed sequences are selected for testing, ensuring a high-quality, balanced dataset for training, validation, and evaluation.
> > > >
> > > > * Programmable RNA Switches
> > > >     We adopt the data generation pipeline described in [r4]. A toehold-switch library comprising 244,000 potential trigger sequences is designed and synthesized, covering the complete genomes of 23 pathogenic viruses, the entire coding regions of 906 human transcription factors, and approximately 10,000 random sequences. Using this synthesized oligo pool, two construct libraries are created to represent the ON and OFF states, and both are transformed into BL21 E. coli. The OFF library includes toehold-switch constructs without triggers, while the ON library contains identical toeholds paired with complementary triggers fused to their respective switches.
> > > >
> > > >   The libraries are sorted into four bins using fluorescence-activated cell sorting (FACS), and the variants in each bin are quantified through next-generation sequencing (NGS) to determine their fluorescence distributions. After quality control, the toehold-switch library consists of 109,067 ON-state measurements, 163,967 OFF-state measurements, and 91,534 ON/OFF paired ratios, where both states are characterized for each switch. ON and OFF data are normalized to a scale of 0 to 1, with ON/OFF ratios normalized to a range of -1 to 1. Following [r4], a stringent quality control process is applied to eliminate artifacts and ensure data reliability. The quality control (QC) framework includes five levels: QC1, QC2, QC3, QC4 and QC5, where QC1 represents the lowest quality and QC5 the highest. Datasets above QC2 are utilized for training, while QC5 is reserved for testing.
> > > >
> > > > * Secondary Structure Prediction
> > > >     We follow the preprocessing steps outlined in the bpRNA-1m dataset [r5].To reduce sequence redundancy and improve dataset diversity, we implement an 80\% sequence-identity threshold and cap the maximum sequence length at 500 nucleotides, following protocols described in the referenced studies. These measures are essential for minimizing overfitting and ensuring that the models are trained on a wide range of genetically diverse samples.
> > > >
> > > >     The dataset is divided into three subsets: a training set (TR0), a validation set (VL0), and a test set (TS0). The splitting process is randomized to eliminate potential biases and ensure an unbiased evaluation of the model’s performance.
> > > >
> > > > [r3] Integration of multiple epigenomic marks improves prediction of variant impact in saturation mutagenesis reporter assay. Human Mutation 2019.
> > > >
> > > > [r4] A deep learning approach to programmable RNA switches. Nature Communications 2020.
> > > >
> > > > [r5] bpRNA: large-scale automated annotation and analysis of RNA secondary structure. Nucleic Acids Research 2018.

---

> > > > > ### Author Response · Authors · 2024-12-03
> > > > > **Finetuning schemes and data nature/quality**
> > > > >
> > > > > * Protein Tasks
> > > > >     We obtain data on thermostability prediction, enzyme commission number prediction and contact map prediction from Saprot [r6]. Following the guidance on github, we download data and place it in the LMDB folder for supervised fine-tuning.
> > > > >
> > > > > * Cross-molecular Tasks
> > > > >     For the enzyme commission number prediction task, to obtain the codon information corresponding to protein sequences, we use the UniProtKB mapping function to convert UniProt IDs into European Nucleotide Archive entries. We then employ the Smith-Waterman algorithm to quickly match the corresponding codon sequences, filtering out all sequences that contained unknown nucleotides or where the number of matched nucleotides is not a multiple of three. For other cross-omics tasks, we adopt the data and settings from [r7].
> > > > >
> > > > > * Enhancer-Promoter Interaction Prediction
> > > > >     We follow the processing of [r8]. We derive the dataset from EPIANN[r9], which includes six cell lines, GM12878, HeLa-S3, IMR90, K562, HUVEC and NHEK. To address the challenge of data imbalance, EPIANN enhanced the representation of positive samples by incorporating the upstream and downstream regions of enhancers. This approach expanded the dataset to include relevant genomic regions by defining extended windows of 3 kbp around enhancers and 2 kbp around promoters, ensuring a more comprehensive capture of the surrounding regulatory landscape.
> > > > >
> > > > > * siRNA Efficiency Prediction
> > > > >     We get the dataset from SAIS[r10]. We use the information of the reference sequence of the target gene, the sense sequence of the target gene, the sense sequence of modified siRNA and the remaining percentage of mRNA after the experiment named `gene_target_seq`, `siRNA_sense_seq`, `modified_siRNA_sense_seq`, and `mRNA_remaining_pct` in dataset from SAIS, respectively.
> > > > >
> > > > > * Antibody-Antigen Neutralizability Prediction
> > > > >     We follow [r11], which provides a minimal dataset specifically designed for this prediction task. This task is based on two datasets: CATNAP[r12], which focuses on HIV, and CoVAbDab[r13], which pertains to SARS-CoV-2.
> > > > >     HIV data is sourced from CATNAP in the Los Alamos HIV Database. Antibody (Ab) and antigen (Ag) sequences are extracted, curated to remove duplicates and missing values, and classified as neutralizing (IC₅₀ < 10 μg/ml) or non-neutralizing (IC₅₀ ≥ 10 μg/ml). Seen and unseen Abs are split, ensuring no overlap between training, validation, and testing sets by excluding similar pairs (BlastP ≥ 90%). Training is conducted on seen Abs, with unseen Abs used for evaluation across 20 random dataset splits.
> > > > >     SARS-CoV-2 Data is collected from CoVAbDab and includes pairwise Ab–Ag instances across variants like Alpha, Beta, Delta, and Omicron. Five sequences per variant and 11 for Omicron are used. Omicron is treated as an unseen Ag, excluded from training but incorporated in relation graphs for transductive learning, enabling the identification of broad-spectrum Abs.
> > > > >
> > > > > * CRISPR Off-Target Prediction Following [r14], we get the off-target dataset, which comprises two different cell types containing 30 sgRNAs. For all 30 sgRNAs, approximately 160,000 possible off-target sites across the entire genome are obtained. Off-target sites are annotated and standardized using the targeting cutting frequency (indel frequency) detected by different off-target detection methods.
> > > > >
> > > > > [r6] Saprot: Protein language modeling with structure-aware vocabulary. ICLR 2024.
> > > > >
> > > > > [r7] Are genomic language models all you need? exploring genomic language models on protein downstream tasks. Bioinformatics 2024.
> > > > >
> > > > > [r8] Predicting enhancer-promoter interactions by deep learning and matching heuristic. Briefings in Bioinformatics 2021.
> > > > >
> > > > > [r9] Modeling enhancer-promoter interactions with attention-based neural networks. bioRxiv 2017.
> > > > >
> > > > > [r10] http://competition.sais.com.cn/competitionDetail/532230/format
> > > > >
> > > > > [r11] Predicting unseen antibodies' neutralizability via adaptive graph neural networks. Nature Machine Intelligence 2022.
> > > > >
> > > > > [r12] CATNAP: a tool to compile, analyze and tally neutralizing antibody panels. Nucleic Acids Research 2015.
> > > > >
> > > > > [r13] CoV-AbDab: the coronavirus antibody database. Bioinformatics 2021.
> > > > >
> > > > > [r14] DeepCRISPR: optimized CRISPR guide RNA design by deep learning. Genome Biology 2018.

---

> > > > > > ### Author Response · Authors · 2024-12-03
> > > > > > **Finetuning schemes and data nature/quality**
> > > > > >
> > > > > > * DNA-Protein Folding Prediction  We query the PDB database using the filenames provided by deepPBD[r15] to obtain the mmCIF files of DNA-protein complexes and get 428 mmCIF files. From the mmCIF files, we extract the coordinates, sequences, and certain bonding information of both DNA and proteins. When encountering modified residues or nucleotides in the mmCIF files, we follow the AlphaFold3[r16] and map these residues or nucleotides to standard amino acids or DNA sequences using SCOP. We set the DNA-protein interface distance threshold to 5Å. Based on this threshold, we derive the DNA-protein interface information. Subsequently, we match the DNA and protein duplex information using the DNA-protein interface and sequence information. Finally, we obtained 683 DNA-protein complexes.
> > > > > >
> > > > > > * RNA-Protein Interaction Prediction
> > > > > >     The dataset is sourced from NPInter2.0[r17], NPInter2.0\_lncRNA[r18], and RPI7317[r19]. The sequences of ncRNAs and proteins are obtained from the NONCODE database[r20], Gencode database[r22], and UniProt database[r21]. The NPInter database integrates new datasets from literature and related resources, with a major focus on data published in recent years. Through a systematic PubMed search using keywords related to RNA interactions, 1270 relevant articles were identified. Verified or processed interaction data were manually extracted, while raw sequencing data were excluded. Binding sites were compared against RefSeq coding genes to remove overlaps with coding regions and cross-checked with NONCODE for ncRNA references. Valid interactions were annotated with standardized IDs (UniProt, RefSeq, NONCODE, etc.) depending on the molecule type.
> > > > > >     Data from external resources like LncRNADisease[r23], which curated 478 experimentally supported lncRNA interactions, were integrated and subjected to the same annotation pipeline. The combined dataset underwent redundancy elimination, aggregating overlapping interactions into single records. NPInter v2.0 thus provides a comprehensive, curated multilevel snapshot of RNA-related interactions.
> > > > > >
> > > > > >
> > > > > > [r15] Geometric deep learning of protein--DNA binding specificity. Nature Methods 2024.
> > > > > >
> > > > > > [r16] Accurate structure prediction of biomolecular interactions with AlphaFold 3. Nature 2024.
> > > > > >
> > > > > > [r17] NPInter v2. 0: an updated database of ncRNA interactions. Nucleic acids research 2014.
> > > > > >
> > > > > > [r18] The bipartite network projection-recommended algorithm for predicting long non-coding RNA-protein interactions. Molecular Therapy-Nucleic Acids 2018.
> > > > > >
> > > > > > [r19] LPI-BLS: Predicting lncRNA--protein interactions with a broad learning system-based stacked ensemble classifier. Neurocomputing 2019.
> > > > > >
> > > > > > [r20] NONCODE v3. 0: integrative annotation of long noncoding RNAs. Nucleic acids research 2012.
> > > > > >
> > > > > > [r21] Update on activities at the Universal Protein Resource (UniProt) in 2013. Nucleic acids research 2012.
> > > > > >
> > > > > > [r22] GENCODE reference annotation for the human and mouse genomes. Nucleic acids research 2019.
> > > > > >
> > > > > > [r23] LncRNADisease: a database for long-non-coding RNA-associated diseases. Nucleic acids research 2012.

---

### Official Review · Reviewer_6UP5 · 2024-11-03

**Soundness:** 4
**Presentation:** 3
**Contribution:** 3
**Rating:** 8
**Confidence:** 4

**Summary:**

The paper presents the performance of eight foundation models of varying design on 17 tasks of varying underlying biological attributes. The authors adapted and trained the models for the

**Strengths:**

The authors grouped various tasks from multiple sources and did exceptional work in clearly detailing the underlying biological importance of the task. They choose multi-omics data. The task not only describes different biological aspects but also varying computational types, as some are classification and some are regression tasks. The training data process was clearly detailed.
They displayed the performance of cross-omics tasks by combining models, a crucial aspect and direction for future research.

**Weaknesses:**

The manuscript does not sufficiently explain the primary criteria for selecting models or tasks. For instance, the authors did not include models like HyenaDNA; it is unclear if this was due to lack of availability, popularity, or suitability. Furthermore, the rationale behind choosing specific tasks, such as those related to gene expression, is underdeveloped. Additional databases like PanglaoDB could have been considered to broaden the task scope. In addition, the task generation.The authors' decision to group cell types also introduces a layer of specificity that might limit the generalizability of conclusions across model families or architectures. If I'm not mistaken the tasks are not shared and if not require some summary statistics to support the conclusions.

Model performance is evaluated based on fine-tuning, but it is unclear whether the observed results are due to the model architecture or the fine-tuning process itself. This could be addressed by performing multiple training runs and reporting the mean and variance to provide a more reliable performance measure. Also, only a single metric is shared, which again limits the ability to generalize.
The term frozen in Table Three is not fully explained. I assume it refers to the use of the model released weights if so for DNABERT the performance of the frozen is better than the non-frozen?. The difference in performance between the frozen and unfrozen should be discussed in the results.

In summary, three main limitations affect this study: (1) the absence of clear criteria for model and task selection, (2) a lack of performance confidence intervals, and (3) insufficient detail in task creation. These weaknesses limit the generalizability of the findings, and hinder reproducibility and application in other contexts. Even the conclusion that “language models outperform naïve supervised models” is hard to ascertain for the

**Questions:**

1.	Detail the criteria for choosing models?
2.	Detail the criteria for creation of the specific tasks (why this DB why not another)?
3.	Elaborate on the performance add confidence intervals and other metrics?
4.	What does the term frozen mean?
5.	Expand on the task creation process or supply the tasks or the summary statistics.

---

> ### Author Response · Authors · 2024-11-24
> **Answer to Reviewer 6UP5**
>
> We sincerely appreciate your thoughtful and constructive feedback. Below, we address each of the identified weaknesses and questions in detail.
>
>
> ### **Question1: Criterial For Choosing Models**
>
> We outline the criteria for selecting models. Through a comprehensive review of comparative studies in the literature, we evaluated models across various omics tasks. For each omics domain, we selected the two top-performing models due to computational constraints. Furthermore, all of the model selections are open-source and reproducible.
>
> - For DNA-related tasks, HyenaDNA, a convolution-based long-sequence model, demonstrated performance comparable to other pretrained DNA language models in DNA generation and understanding tasks. However, based on findings from sources such as BEND [r1] and Genomics-FM [r2], we identified DNABERT2 and NTv2-multispecies as the models that achieved the best performance across most tasks. Therefore, we selected DNABERT2 and NTv2-M for DNA omics.
> - In RNA omics, taking into account the excellent performance and the complete open-source nature, we referred to evaluations from the BEACON [r4] benchmark and selected RNA-FM and BEACON-B as the top-performing models.
> - For proteomics, results from PEER [r5] and ProteinGym [r6] led us to initially focus on the ESM (Evolutionary Scale Modeling) family of models. Specifically, we selected ESM-1b for testing. Following the release of updated versions in the ESM series, we included ESM-2 as an additional test model.
>
> At the time of our experiments, LucaOne was the only available multi-omics language model. Consequently, LucaOne was chosen for tasks involving multi-omics.
>
> [r1] BEND: Benchmarking DNA Language Models on biologically meaningful tasks. ICLR 2024.
>
> [r2] Genomics-FM: Universal Foundation Model for Versatile and Data-Efficient Functional Genomic Analysis. bioRxiv 2024.
>
> [r3] BEACON: Benchmark for Comprehensive RNA Tasks and Language Models. NeurIPS 2024.
>
> [r4] PEER: A Comprehensive and Multi-Task Benchmark for Protein Sequence Understanding. NeurIPS 2022.
>
> [r5] ProteinGym: Large-Scale Benchmarks for Protein Design and Fitness Prediction. NeurIPS 2023.

---

> > ### Author Response · Authors · 2024-11-24
> > **Answer to Reviewer 6UP5**
> >
> > ### **Question2: Criteria For Selecting Tasks**
> >
> > We curated a comprehensive collection of tasks encompassing diverse molecular types and enlisted evaluations from several biology professors and PhD students. From these assessments, we selected a representative subset of tasks to establish the first multi-omics benchmark spanning DNA, RNA, and proteins.
> >
> > - As shown in Table 1 of our paper, the tasks were sourced from high-impact conferences, journals, and competitions, emphasizing literature with high citation counts. Many of these tasks have already been used to evaluate the performance of biological language models specific to their respective omics. For instance, Gene Expression data originates from Cell Reports, Enhancer Activity Prediction from Nature Genetics, and APA Isoform Prediction from Cell. Similarly, tasks such as Programmable RNA Switches and Secondary Structure Prediction derive from Nature Communications and Nucleic Acids Research, respectively, while others like Thermostability Prediction and Contact Map Prediction are sourced from NeurIPS and BMC Bioinformatics. The selected tasks span both single-omics and cross-omics domains and broadly encompass the aspects of structure, function, and engineering.
> >
> > - Notably, for DNA tasks, we included Gene Expression Prediction, which forecasts the cross-tissue expression levels of genes and transcription factors (TFs), shedding light on regulatory networks underlying cell states and tissue-specific genomic functions. Enhancer Activity Prediction, on the other hand, analyzes DNA sequences to predict enhancer activity on specific promoters, revealing how regulatory signals drive transcriptional specificity in different cell types. These tasks also vary significantly in sequence lengths—Gene Expression tasks use sequences of 6000 bp, while Enhancer Activity Prediction involves sequences of 249 bp for evaluation of model performance across varying DNA sequence lengths. For the other expression dataset, PanglaoDB focuses on cell type identification and classification in single-cell RNA sequencing (scRNA-seq) datasets and provides curated lists of marker genes for specific cell types across tissues in humans and mice. However, it is mainly used in single-cell research with non-sequence data, for instance, scBERT [r1]. In our study, the gene expression task utilizes the Xpresso dataset from the Epigenomics Roadmap Consortium [r2], which is at the bulk level and emphasizes the regulatory effects of cell type-specific non-coding regions on gene expression, as well as inputs into the model as the sequence data.
> >
> > - Looking ahead, we aim to expand the benchmark by incorporating additional high-impact tasks that span multiple omics and broaden the scope of structural and functional predictions, driving innovation in bioinformatics and computational biology.
> >
> > [r1] scBERT as a large-scale pretrained deep language model for cell type annotation of single-cell RNA-seq data. Nature Machine Intelligence 2022.
> >
> > [r2] Integrative analysis of 111 reference human epigenomes. Nature 2015.

---

> > > ### Author Response · Authors · 2024-11-24
> > > **Answer to Reviewer 6UP5**
> > >
> > > ### **Question3: Confidence Interval of The Experiment**
> > >
> > > Thank you for your valuable suggestions. To address your concerns, we conducted additional experiments to evaluate the impact of varying random seeds under identical optimal training hyperparameters. By reporting the mean and standard deviation of the results, we aim to provide greater confidence in our results. Specifically, we employed three random seeds for these experiments. Due to limitations in computational resources and time, we prioritized completing the full set of single-omics (Table 1 and 2) and multi-molecular  (Table 3 and 4)  tasks with the backbone frozen and naive supervised model and the results are shown in the following table.
> > >
> > > **More seeds against one seed experiments for single-omics results**
> > >
> > > Table 1: Results of different models on single-molecular tasks evaluated with three random seeds. Mean (std) is reported for each experiment.
> > >
> > > ||DNA|||RNA|||Protein|||
> > > |:-------------------------------:|:-----:|:-------:|:------:|:-----:|:-----:|:-----:|:-------:|:------------:|:-----:|
> > > |Model/Task|GE|EA(Dev)|EA(Hk)|APA|PRS|SSP|Cont|Ther|EC|
> > > |Metric|R^2|PCC|PCC|R^2|R^2|F1|P@ L/5|Speraman's p|fmax|
> > > |Naive Supervised Model||||||||||
> > > |CNN|37.38(2.59)|66.03(0.15)|74.48(0.21)|50.93(0.17)|45.40(0.66)|49.95(0.82)||||
> > > |ResNet|35.73(2.67)|67.29(0.15)|75.92(0.10)|56.45(0.94)|55.21(0.28)|57.26(3.14)||||
> > > |LSTM|41.63(0.29)|68.82(0.14)|76.83(0.22)|67.03(0.86)|55.45(0.71)|58.61(0.21)||||
> > > |Pretrained Omics Language Model (Frozen)||||||||||
> > > |DNABERT2|14.19(0.67)|38.46(0.02)|40.45(1.13)|38.68(1.77)|20.37(0.37)|13.56(1.66)|7.09(1.41)|59.07(1.66)|47.18(1.41)|
> > > |NTv2|13.05(1.54)|30.80(1.25)|31.70(2.43)|36.25(5.74)|25.22(1.91)|14.79(1.02)|7.26(0.17)|60.08(1.02)|40.47(0.17)|
> > > |RNA-FM|37.18(0.20)|36.27(0.04)|37.87(0.26)|30.40(2.42)|20.21(0.15)|64.64(0.64)|1.71(1.76)|56.28(0.64)|29.56(1.76)|
> > > |BEACON-B|23.16(0.53)|34.47(0.10)|38.55(0.07)|35.65(9.37)|25.81(1.01)|58.51(0.59)|10.74(1.63)|57.52(0.59)|37.19(1.63)|
> > > |ESM-1b|30.96(4.13)|51.78(0.86)|63.48(0.49)|52.48(4.48)|51.89(0.36)|30.52(0.83)|38.65(0.11)|69.43(0.83)|88.08(0.11)|
> > > |ESM2|31.94(3.08)|45.31(1.92)|56.87(3.70)|36.37(7.86)|35.57(0.19)|44.65(2.90)|43.32(0.49)|65.47(2.90)|77.60(0.49)|
> > > |LucaOne|41.25(0.57)|46.86(0.01)|51.44(0.01)|40.38(0.61)|38.85(0.11)|57.66(0.30)|4.00(0.06)|66.66(0.30)|73.69(0.06)|
> > >
> > > Table 2: Results of different models on single-molecular tasks evaluated with one random seed.
> > >
> > > |                                          |  DNA  |         |        |  RNA  |       |       | Protein |              |        |
> > > |:----------------------------------------:|:-----:|:-------:|:------:|:-----:|:-----:|:-----:|:-------:|:------------:|:------:|
> > > |                Model/Task                |   GE  | EA(Dev) | EA(Hk) |  APA  |  PRS  |  SSP  |   Cont  |     Ther     |   EC   |
> > > |                  Metric                  |  R^2  |   PCC   |   PCC  |  R^2  |  R^2  |   F1  |  P@ L/5 | Speraman's p |  fmax  |
> > > |          Naive Supervised Model          |       |         |        |       |       |       |         |              |        |
> > > |                    CNN                   | 34.52 |  66.08  |  74.29 | 50.93 |  45.2 | 49.95 |         |              |        |
> > > |                  ResNet                  | 38.65 |  67.41  |  75.84 | 56.45 | 55.33 | 57.26 |         |              |        |
> > > |                   LSTM                   | 41.34 |  68.93  |  77.02 | 67.03 | 56.54 | 58.61 |         |              |        |
> > > | Pretrained Omics Language Model (Frozen) |       |         |        |       |       |       |         |              |        |
> > > |                 DNABERT2                 | 13.82 |  39.44  |  41.75 | 40.48 | 19.99 | 13.51 |  11.24  |     60.94    |  47.36 |
> > > |                   NTv2                   | 13.78 |  29.36  |  28.89 | 30.86 | 23.09 | 14.72 |   7.48  |     60.95    |  40.57 |
> > > |                  RNA-FM                  | 37.15 |  36.31  |  37.86 | 32.88 | 20.05 | 64.43 |   2.07  |     56.80     |  29.93 |
> > > |                 BEACON-B                 | 23.61 |  34.57  |  38.63 | 41.21 | 25.73 | 58.73 |  12.15  |     57.18    |  39.08 |
> > > |                  ESM-1b                  | 28.97 |  51.71  |  63.31 | 53.11 | 52.29 | 31.25 |  39.09  |     69.83    |  88.17 |
> > > |                   ESM2                   | 28.99 |  43.27  |  54.73 | 27.38 | 35.51 | 44.08 |  43.34  |     63.02    |  77.80  |
> > > |                  LucaOne                 | 41.48 |  46.86  |  51.44 | 40.11 | 38.95 | 56.86 |   3.84  |    66.33    | 73.65 |

---

> > > > ### Author Response · Authors · 2024-11-24
> > > > **Answer to Reviewer 6UP5**
> > > >
> > > > **More seeds against one seed experiments for multi-omics results**
> > > >
> > > > Table 3: Results of different models on homo-omics multi-molecules evaluated with three random seeds. Mean (std) is reported for each experiment.
> > > >
> > > > |Model/Task|EPI|Model/Task|AAN|Model/Task|siRNA|
> > > > |------------------------------------------|---------|-------------------|---------|-------------------|-----------------|
> > > > |Metric|MCC (%)|Metric|MCC (%)|Metric|Mixed Score (%)|
> > > > |Naive Supervised Model||||||
> > > > |CNN|27.19(1.88)|CNN|39.36(5.71)|CNN|57.35(1.10)|
> > > > |ResNet|56.26(0.66)|ResNet|38.79(6.79)|ResNet|52.67(13.07)|
> > > > |LSTM|58.78(2.80)|LSTM|38.39(1.55)|LSTM|47.85(1.50)|
> > > > |Pretrained Omics Language Model (Frozen)||||||
> > > > |DNABERT2+NTv2|13.17(1.30)|ESM1b+ESM2|43.79(1.39)|BEACON-B+RNA-FM|49.67(0.17)|
> > > > |DNABERT2+DNABERT2|13.86(3.90)|ESM1b+ESM1b|48.11(0.08)|BEACON-B+BEACON-B|49.60(0.39)|
> > > > |NTv2+DNABERT2|12.76(6.32)|ESM2+ESM1b|42.54(1.12)|RNA-FM+BEACON-B|49.76(0.26)|
> > > > |NTv2+NTv2|11.39(1.46)|ESM2+ESM2|38.42(0.95)|RNA-FM+RNA-FM|49.50(0.01)|
> > > > |LucaOne|15.84(0.61)|LucaOne|25.20(0.44)|LucaOne|50.04(0.29)|
> > > > |||||||
> > > > |Model/Task|RPI|Model/Task|CRI-Off|Model/Task|DPF|
> > > > |Metric|MCC (%)|Metric|SCC (%)|Metric|LDDT (%)|
> > > > |Naive Supervised Model||||||
> > > > |CNN|86.24(0.27)|CNN|10.19(1.06)|CNN|32.84(1.63)|
> > > > |ResNet|86.82(0.55)|ResNet|6.87(3.36)|ResNet|33.62(0.61)|
> > > > |LSTM|87.09(0.76)|LSTM|9.39(1.53)|LSTM|31.02(1.29)|
> > > > |Pretrained Omics Language Model (Frozen)||||||
> > > > |ESM1b+RNA-FM|84.16(0.18)|RNA-FM+NTv2|6.07(0.20)|NTv2+ESM-1b|39.15(1.29)|
> > > > |ESM2+RNA-FM|82.57(0.40)|RNA-FM+DNABERT2|1.87(1.96)|DNABERT2+ESM-1b|39.15(0.74)|
> > > > |ESM1b+BEACON-B|85.57(0.44)|BEACON-B+NTv2|3.11(1.72)|NTv2+ESM-2|42.47(0.93)|
> > > > |ESM2+BEACON-B|83.37(0.58)|BEACON-B+DNABERT2|3.18(1.76)|DNABERT2+ESM-2|42.69(0.71)|
> > > > |LucaOne|77.80(0.13)|LucaOne|9.17(0.69)|LucaOne|35.94(3.17)|

---

> > > > > ### Author Response · Authors · 2024-11-24
> > > > > **Answer to Reviewer 6UP5**
> > > > >
> > > > > Table 4: Results of different models on homo-omics multi-molecules evaluated with one random seed.
> > > > >
> > > > > |                Model/Task                |   EPI   |     Model/Task    |   AAN   |     Model/Task    |      siRNA      |
> > > > > |:----------------------------------------:|:-------:|:-----------------:|:-------:|:-----------------:|:---------------:|
> > > > > |                  Metric                  | MCC (%) |       Metric      | MCC (%) |       Metric      | Mixed Score (%) |
> > > > > |          Naive Supervised Model          |         |                   |         |                   |                 |
> > > > > |                    CNN                   |  25.04  |        CNN        |  39.08  |        CNN        |      56.41      |
> > > > > |                  ResNet                  |  56.76  |       ResNet      |  45.79  |       ResNet      |      61.74      |
> > > > > |                   LSTM                   |  58.47  |        LSTM       |  39.73  |        LSTM       |      48.69      |
> > > > > | Pretrained Omics Language Model (Frozen) |         |                   |         |                   |                 |
> > > > > |               DNABERT2+NTv2              |  11.67  |     ESM1b+ESM2    |  44.31  |  BEACON-B+RNA-FM  |      49.86      |
> > > > > |             DNABERT2+DNABERT2            |   10.60  |    ESM1b+ESM1b    |  48.09  | BEACON-B+BEACON-B |      49.73      |
> > > > > |               NTv2+DNABERT2              |   6.47  |     ESM2+ESM1b    |   42.8  |  RNA-FM+BEACON-B  |      50.05      |
> > > > > |                 NTv2+NTv2                |  13.06  |     ESM2+ESM2     |  39.42  |   RNA-FM+RNA-FM   |      49.48      |
> > > > > |                  LucaOne                 |  15.16  |      LucaOne      |  25.55  |      LucaOne      |      50.13      |
> > > > > |                                          |         |                   |         |                   |                 |
> > > > > |                Model/Task                |   RPI   |     Model/Task    | CRI-Off |     Model/Task    |       DPF       |
> > > > > |                  Metric                  | MCC (%) |       Metric      | SCC (%) |       Metric      |     LDDT (%)    |
> > > > > |          Naive Supervised Model          |         |                   |         |                   |                 |
> > > > > |                    CNN                   |  86.25  |        CNN        |  10.86  |        CNN        |      34.64      |
> > > > > |                  ResNet                  |  87.39  |       ResNet      |   8.90  |       ResNet      |      33.14      |
> > > > > |                   LSTM                   |  87.83  |        LSTM       |   7.63  |        LSTM       |      32.09      |
> > > > > | Pretrained Omics Language Model (Frozen) |         |                   |         |                   |                 |
> > > > > |               ESM1b+RNA-FM               |  83.96  |    RNA-FM+NTv2    |   5.89  |    NTv2+ESM-1b    |      40.59      |
> > > > > |                ESM2+RNA-FM               |  82.83  |  RNA-FM+DNABERT2  |   3.87  |  DNABERT2+ESM-1b  |      39.90      |
> > > > > |              ESM1b+BEACON-B              |  85.64  |   BEACON-B+NTv2   |   4.7   |     NTv2+ESM-2    |      42.53      |
> > > > > |               ESM2+BEACON-B              |  84.01  | BEACON-B+DNABERT2 |   3.14  |   DNABERT2+ESM-2  |      43.39      |
> > > > > |                  LucaOne                 |   77.90  |      LucaOne      |   8.42  |      LucaOne      |      32.65      |

---

> > > > > > ### Author Response · Authors · 2024-11-24
> > > > > > **Answer to Reviewer 6UP5**
> > > > > >
> > > > > > The experimental results demonstrate:
> > > > > >
> > > > > > * The fine-tuning of the frozen pretrained Omics Language Model exhibits a low standard deviation across multiple random seeds, and the mean of the multi-seed experiments closely matches the results of single-seed experiments.
> > > > > > * For the naive supervised models, the multi-seed mean results align closely with single-seed results across all experiments, except for the ResNet experiment.
> > > > > >
> > > > > > Regarding the diverse and complex tasks covered in the paper:
> > > > > > Our work spans three categories—single-omics, cross-molecular, and multi-molecular—comprising a total of 17 tasks. These tasks address a broad range of objectives, including structure, function, and engineering, and encompass various task types, such as single-label regression, multi-label regression, and multi-label classification. Additionally, the size of the test sets varies significantly, ranging from 40 to 49,755 samples. Given this complexity, we adhered to the evaluation metrics recommended in the original dataset publications to assess model performance comprehensively.

---

> > > > > > > ### Comment · Reviewer_6UP5 · 2024-11-26
> > > > > > >
> > > > > > > Than you for addressing my comments

---

> > > > ### Comment · Reviewer_6UP5 · 2024-11-26
> > > >
> > > > Thank you for addressing this concern. This is hard work indeed. I would suggest that due to a large number of numbers, you can focus the reader by marking the values where the sd is of the same scale as the mean, such as the RNA-FM or mark the best performance (if applicable, given the sd). Again great work.

---

> > ### Comment · Reviewer_6UP5 · 2024-11-26
> > **≈**
> >
> > Thank you for addressing this concern

---

> ### Author Response · Authors · 2024-11-24
> **Answer to Reviewer 6UP5**
>
> ### **Question4: Explanation of The Term Frozen**
>
> Broadly, we froze the released weights of the model's backbone while training classification heads for all tasks across all models. For cross-omics tasks, such as fine-tuning protein language models (ESM2) on nucleotide sequence tasks (APA Isoform Prediction), both the word embedding layer and the classification heads were trained.
> Following the experimental setups used in BEND, PEER and computer vision studies, we adopted a fine-tuning approach with the backbone frozen. This approach serves two purposes:
>
> - evaluating the quality of the representations learned during pretraining
> - exploring whether fine-tuning on downstream tasks disrupts the knowledge acquired during pretraining.
>
> In tasks like Contact Map Prediction and Thermostability Prediction, where DNABERT2 with a frozen backbone performed better than full-parameter fine-tuning, we attribute this to the latter potentially overwriting or forgetting knowledge gained during pretraining, suggesting that full-parameter fine-tuning may, under certain conditions, lead to catastrophic forgetting.
>
>
> ### **Question5: Detailed Preprocessing for Each Task**
>
> Thank you for your advice! The detailed task creation process has been included in the manuscript `APPENDIX A.4` of the revised version in `green`.
>
> * Gene Expression
>     We adopt the data processing methodology from Xpresso[r1]. Human gene expression data comes from the Epigenomics Roadmap Consortium, which provides normalized RNA-seq values for protein-coding mRNAs across 56 tissues and cell lines.
>
>     Due to the large number of parameters in biological language models and the memory limitations of A100 GPUs, our experiments show that trimming sequence lengths to 6000 bp ensures compatibility with all models for processing input sequences. By inputting consecutive 6000 bp nucleotide fragments from different positions in the processed sequences into the Xpresso model, we identify that the sequence indexed from position 7000 to 12999 (length 6000 bp) achieves optimal test performance. This segment contains the most information related to gene expression levels.
>
>     For training, we use the 6000 bp nucleotide sequence indexed from position 7000 to 12999 as input and the expression data for 56 tissues as labels. The train, validation, and test dataset splits follow the methodology used in Xpresso.
>
> * Enhancer Activity Prediction
>     We follow the processing procedure described in [r2]. The data includes sequence information and transcriptional activity metrics for both Drosophila and humans, encompassing developmental and housekeeping transcriptional activity levels.
>
>     We use downloaded sequences of 249 bp in length, along with `Dev_log2_enrichment_scaled` and `Hk_log2_enrichment_scaled`, which respectively represent developmental and housekeeping transcriptional activity information. The dataset is divided into training, validation, and test sets according to the method outlined in [r2].
>
>
> * APA Isoform Prediction
>     The preparation for IPA isoform analysis begins by filtering raw sequencing reads from all MPRAs[r3] to retain only high-quality, full-length RNA sequences. These reads are grouped based on the randomized regions located upstream of the proximal polyadenylation site (pPAS), forming a dictionary of sequence variants for each library. To expand this dictionary, sequencing is also performed on the plasmid library, capturing members that lack expression of a distal isoform. RNA reads are then matched to dictionary entries by identifying the upstream region with the shortest Hamming distance.
>
>     Polyadenylation cleavage sites are determined for each mapped read by detecting the presence of a Poly-A tail. The cleavage positions are recorded as vectors associated with individual sequence variants, including a specific position for reads mapping to non-random distal sites. The dataset generated from this process consists of a dictionary of distinct sequence variants paired with vectors of cleavage position counts. A final filtering step ensures data quality by discarding sequences supported by fewer than 10–20 unique UMI RNA reads or those containing over 75\% A-nucleotides within a 12–20 bp region, which could indicate internal priming artifacts.
>
> [r1] Predicting mrna abundance directly from genomic sequence using deep convolutional neural networks. Cell Reports 2020.
>
> [r2] DeepSTARR predicts enhancer activity from DNA sequence and enables the de novo design of synthetic enhancers. Nature Genetics 2022.
>
> [r3] Integration of multiple epigenomic marks improves prediction of variant impact in saturation mutagenesis reporter assay. Human Mutation 2019.

---

> ### Author Response · Authors · 2024-11-24
> **Answer to Reviewer 6UP5**
>
> We process data from 12 random 3' UTR libraries. 9 among the 12 libraries are used for training and 3 held out  (the 3 held-out libraries were excluded from the current analysis). To construct a balanced test set, sequences from each library are first shuffled independently according to their read counts. These shuffled sequences are then merged using a round-robin approach, selecting one sequence from each library at a time in descending order of read count. This strategy ensures that the test set contains an even representation of high-read count sequences across all libraries. The remaining sequences are appended to the beginning of the combined library, and the training set is further shuffled to enhance randomness. For benchmarking purposes, the top 10\% of high-read count sequences are prioritized. Among these, the most abundantly expressed sequences are selected for testing, ensuring a high-quality, balanced dataset for training, validation, and evaluation.
>
> * Programmable RNA Switches
>     We adopt the data generation pipeline described in [r4]. A toehold-switch library comprising 244,000 potential trigger sequences is designed and synthesized, covering the complete genomes of 23 pathogenic viruses, the entire coding regions of 906 human transcription factors, and approximately 10,000 random sequences. Using this synthesized oligo pool, two construct libraries are created to represent the ON and OFF states, and both are transformed into BL21 E. coli. The OFF library includes toehold-switch constructs without triggers, while the ON library contains identical toeholds paired with complementary triggers fused to their respective switches.
>
>     The libraries are sorted into four bins using fluorescence-activated cell sorting (FACS), and the variants in each bin are quantified through next-generation sequencing (NGS) to determine their fluorescence distributions. After quality control, the toehold-switch library consists of 109,067 ON-state measurements, 163,967 OFF-state measurements, and 91,534 ON/OFF paired ratios, where both states are characterized for each switch. ON and OFF data are normalized to a scale of 0 to 1, with ON/OFF ratios normalized to a range of -1 to 1. Following [r4], a stringent quality control process is applied to eliminate artifacts and ensure data reliability. The quality control (QC) framework includes five levels: QC1, QC2, QC3, QC4 and QC5, where QC1 represents the lowest quality and QC5 the highest. Datasets above QC2 are utilized for training, while QC5 is reserved for testing.
>
> * Secondary Structure Prediction
>     We follow the preprocessing steps outlined in the bpRNA-1m dataset [r5].To reduce sequence redundancy and improve dataset diversity, we implement an 80\% sequence-identity threshold and cap the maximum sequence length at 500 nucleotides, following protocols described in the referenced studies. These measures are essential for minimizing overfitting and ensuring that the models are trained on a wide range of genetically diverse samples.
>
>     The dataset is divided into three subsets: a training set (TR0), a validation set (VL0), and a test set (TS0). The splitting process is randomized to eliminate potential biases and ensure an unbiased evaluation of the model’s performance.
>
> * Protein Tasks
>     We obtain data on thermostability prediction, enzyme commission number prediction and contact map prediction from Saprot [r6]. Following the guidance on github, we download data and place it in the LMDB folder for supervised fine-tuning.
>
> * Cross-molecular Tasks
>     For the enzyme commission number prediction task, to obtain the codon information corresponding to protein sequences, we use the UniProtKB mapping function to convert UniProt IDs into European Nucleotide Archive entries. We then employ the Smith-Waterman algorithm to quickly match the corresponding codon sequences, filtering out all sequences that contained unknown nucleotides or where the number of matched nucleotides is not a multiple of three. For other cross-omics tasks, we adopt the data and settings from [r7].
>
> [r4] A deep learning approach to programmable RNA switches. Nature Communications 2020.
>
> [r5] bpRNA: large-scale automated annotation and analysis of RNA secondary structure. Nucleic Acids Research 2018.
>
> [r6] Saprot: Protein language modeling with structure-aware vocabulary. ICLR 2024.
>
> [r7] Are genomic language models all you need? exploring genomic language models on protein downstream tasks. Bioinformatics 2024.

---

> ### Author Response · Authors · 2024-11-24
> **Answer to Reviewer 6UP5**
>
> * Enhancer-Promoter Interaction Prediction
>     We follow the processing of [r8]. We derive the dataset from EPIANN[r9], which includes six cell lines, GM12878, HeLa-S3, IMR90, K562, HUVEC and NHEK. To address the challenge of data imbalance, EPIANN enhanced the representation of positive samples by incorporating the upstream and downstream regions of enhancers. This approach expanded the dataset to include relevant genomic regions by defining extended windows of 3 kbp around enhancers and 2 kbp around promoters, ensuring a more comprehensive capture of the surrounding regulatory landscape.
>
> * siRNA Efficiency Prediction
>     We get the dataset from SAIS[r10]. We use the information of the reference sequence of the target gene, the sense sequence of the target gene, the sense sequence of modified siRNA and the remaining percentage of mRNA after the experiment named `gene_target_seq`, `siRNA_sense_seq`, `modified_siRNA_sense_seq`, and `mRNA_remaining_pct` in dataset from SAIS, respectively.
>
> * Antibody-Antigen Neutralizability Prediction
>     We follow [r11], which provides a minimal dataset specifically designed for this prediction task. This task is based on two datasets: CATNAP[r12], which focuses on HIV, and CoVAbDab[r13], which pertains to SARS-CoV-2.
>     HIV data is sourced from CATNAP in the Los Alamos HIV Database. Antibody (Ab) and antigen (Ag) sequences are extracted, curated to remove duplicates and missing values, and classified as neutralizing (IC₅₀ < 10 μg/ml) or non-neutralizing (IC₅₀ ≥ 10 μg/ml). Seen and unseen Abs are split, ensuring no overlap between training, validation, and testing sets by excluding similar pairs (BlastP ≥ 90%). Training is conducted on seen Abs, with unseen Abs used for evaluation across 20 random dataset splits.
>     SARS-CoV-2 Data is collected from CoVAbDab and includes pairwise Ab–Ag instances across variants like Alpha, Beta, Delta, and Omicron. Five sequences per variant and 11 for Omicron are used. Omicron is treated as an unseen Ag, excluded from training but incorporated in relation graphs for transductive learning, enabling the identification of broad-spectrum Abs.
>
> * RNA-Protein Interaction Prediction
>     The dataset is sourced from NPInter2.0[r17], NPInter2.0\_lncRNA[r18], and RPI7317[r19]. The sequences of ncRNAs and proteins are obtained from the NONCODE database[r20], Gencode database[r22], and UniProt database[r21]. The NPInter database integrates new datasets from literature and related resources, with a major focus on data published in recent years. Through a systematic PubMed search using keywords related to RNA interactions, 1270 relevant articles were identified. Verified or processed interaction data were manually extracted, while raw sequencing data were excluded. Binding sites were compared against RefSeq coding genes to remove overlaps with coding regions and cross-checked with NONCODE for ncRNA references. Valid interactions were annotated with standardized IDs (UniProt, RefSeq, NONCODE, etc.) depending on the molecule type.
>     Data from external resources like LncRNADisease[r23], which curated 478 experimentally supported lncRNA interactions, were integrated and subjected to the same annotation pipeline. The combined dataset underwent redundancy elimination, aggregating overlapping interactions into single records. NPInter v2.0 thus provides a comprehensive, curated multilevel snapshot of RNA-related interactions.
>
>
>
> [r8] Predicting enhancer-promoter interactions by deep learning and matching heuristic. Briefings in Bioinformatics 2021.
>
> [r9] Modeling enhancer-promoter interactions with attention-based neural networks. bioRxiv 2017.
>
> [r10] http://competition.sais.com.cn/competitionDetail/532230/format
>
> [r11] Predicting unseen antibodies' neutralizability via adaptive graph neural networks. Nature Machine Intelligence 2022.
>
> [r12] CATNAP: a tool to compile, analyze and tally neutralizing antibody panels. Nucleic Acids Research 2015.
>
> [r13] CoV-AbDab: the coronavirus antibody database. Bioinformatics 2021.
>
> [r17] NPInter v2. 0: an updated database of ncRNA interactions. Nucleic acids research 2014.
>
> [r18] The bipartite network projection-recommended algorithm for predicting long non-coding RNA-protein interactions. Molecular Therapy-Nucleic Acids 2018.
>
> [r19] LPI-BLS: Predicting lncRNA--protein interactions with a broad learning system-based stacked ensemble classifier. Neurocomputing 2019.
>
> [r20] NONCODE v3. 0: integrative annotation of long noncoding RNAs. Nucleic acids research 2012.
>
> [r21] Update on activities at the Universal Protein Resource (UniProt) in 2013. Nucleic acids research 2012.
>
> [r22] GENCODE reference annotation for the human and mouse genomes. Nucleic acids research 2019.
>
> [r23] LncRNADisease: a database for long-non-coding RNA-associated diseases. Nucleic acids research 2012.

---

> > ### Author Response · Authors · 2024-11-24
> > **Answer to Reviewer 6UP5**
> >
> > * CRISPR Off-Target Prediction
> >     Following [r14], we get the off-target dataset, which comprises two different cell types containing 30 sgRNAs. For all 30 sgRNAs, approximately 160,000 possible off-target sites across the entire genome are obtained. Off-target sites are annotated and standardized using the targeting cutting frequency (indel frequency) detected by different off-target detection methods.
> >
> > * DNA-Protein Folding Prediction
> >     We query the PDB database using the filenames provided by deepPBD[r15] to obtain the mmCIF files of DNA-protein complexes and get 428 mmCIF files. From the mmCIF files, we extract the coordinates, sequences, and certain bonding information of both DNA and proteins. When encountering modified residues or nucleotides in the mmCIF files, we follow the AlphaFold3[r16] and map these residues or nucleotides to standard amino acids or DNA sequences using SCOP. We set the DNA-protein interface distance threshold to 5Å. Based on this threshold, we derive the DNA-protein interface information. Subsequently, we match the DNA and protein duplex information using the DNA-protein interface and sequence information. Finally, we obtained 683 DNA-protein complexes.
> >
> > [r14] DeepCRISPR: optimized CRISPR guide RNA design by deep learning. Genome Biology 2018.
> >
> > [r15] Geometric deep learning of protein--DNA binding specificity. Nature Methods 2024.
> >
> > [r16] Accurate structure prediction of biomolecular interactions with AlphaFold 3. Nature 2024.

---

> ### Comment · Reviewer_6UP5 · 2024-11-26
>
> Great work thanks for the detailed explanations and re-runs I have decided to increase the score. I think that the multiple runs presented and the more thorough explanations of the models and tasks make this work more valuable.

---

> > ### Author Response · Authors · 2024-11-27
> > **Thank Reviewer 6UP5 so much**
> >
> > We sincerely appreciate your recognition of our efforts to provide detailed explanations and additional experiments. Your constructive insights have been invaluable in improving the clarity and robustness of our work. We’re grateful for your support and encouragement!

---

### Meta-Review · Area_Chair_EgCW · 2024-12-21

**Metareview:**

This paper presents a comprehensive multi-omics benchmark, called COMET (Benchmark for Biological COmprehensive Multi-omics Evaluation Tasks and Language Models).
The benchmark is designed to evaluate models for diverse single/cross/multi-omics tasks.
The reviewers note that the presented benchmark is comprehensive and has the potential to provide useful resources for the evaluation and optimization of (multi-omics) language models.
However, more comprehensive benchmarking results across different models, additional and deeper insights into the major factors of the various models leading to their performance assessment outcomes, further rationale/explanation regarding the benchmark and assessments (e.g., for model selection, task design, benchmark construction, and providing additional context of the current work in relation to other recent papers may be required to further strengthen the current study.

**Additional Comments On Reviewer Discussion:**

The authors have provide extensive additional explanations, justification, and additional results, which have addressed some of the reviewers' initial concerns.
There have been some disagreement between the authors and reviewers regarding the main scope & contribution of the work (e.g., regarding methodological contribution) as well as what would be required in good/useful benchmark papers.
These have been taken into consideration in the AC's recommendation.
While the AC sees the potential value of the current work, there appears to be room for further improvement as noted above and the AC expects that the manuscript can benefit from a major revision to address these points.

---

### Decision · Program_Chairs · 2025-01-22

Reject